# Neuropathologist-level integrated classification of adult-type diffuse gliomas using deep learning from whole-slide pathological images

Weiwei Wang[1,11], Yuanshen Zhao[2,3,11], Lianghong Teng[4,11], Jing Yan[5,11], Yang Guo[6], Yuning Qiu[7], Yuchen Ji[7], Bin Yu[7], Dongling Pei[7], Wenchao Duan[7], Minkai Wang[7], Li Wang[1], Jingxian Duan[2,3], Qiuchang Sun[2,3], Shengnan Wang[4], Huanli Duan[4], Chen Sun[7], Yu Guo[7], Lin Luo[7], Zhixuan Guo[7], Fangzhan Guan[7], Zilong Wang[7], Aoqi Xing[7], Zhongyi Liu[7], Hongyan Zhang[1], Li Cui[1], Lan Zhang[1], Guozhong Jiang[1], Dongming Yan[7], Xianzhi Liu[7], Hairong Zheng[2,3,8,9], Dong Liang[2,3,8,10], Wencai Li[1] ✉, Zhi-Cheng Li[2,3,8,10] ✉ & Zhenyu Zhang[7] ✉

Current diagnosis of glioma types requires combining both histological features and molecular characteristics, which is an expensive and time-consuming procedure. Determining the tumor types directly from whole-slide images (WSIs) is of great value for glioma diagnosis. This study presents an integrated diagnosis model for automatic classification of diffuse gliomas from annotation-free standard WSIs. Our model is developed on a training cohort ($n = 1362$) and a validation cohort ($n = 340$), and tested on an internal testing cohort ($n = 289$) and two external cohorts ($n = 305$ and 328, respectively). The model can learn imaging features containing both pathological morphology and underlying biological clues to achieve the integrated diagnosis. Our model achieves high performance with area under receiver operator curve all above 0.90 in classifying major tumor types, in identifying tumor grades within type, and especially in distinguishing tumor genotypes with shared histological features. This integrated diagnosis model has the potential to be used in clinical scenarios for automated and unbiased classification of adult-type diffuse gliomas.

Diffuse gliomas, which account for the majority of malignant brain tumors in adults, comprise astrocytoma, oligodendroglioma, and glioblastoma[1,2]. The prognosis of diffuse gliomas varies, with median survival being 60–119 months in oligodendroglioma, 18–36 months in astrocytoma, and 8 months in glioblastoma[1]. The fifth edition of the World Health Organization (WHO) Classification of Tumors of the Central Nervous System (CNS) released in 2021 has categorized adult-type diffuse gliomas into three types: (1) astrocytoma, isocitrate

dehydrogenase (IDH)-mutant, (2) oligodendroglioma, IDH-mutant, and 1p/19q-codeleted, and (3) glioblastoma, IDH-wildtype (short for A, O, and GBM)[2]. This newest edition has combined not only established histological diagnosis but also molecular markers for achieving an integrated classification of adult diffuse gliomas[2,3].

In a clinical scenario, integrated diagnosis by combining histological and molecular features of glioma is a time-consuming and laborious procedure, as well as an economically expensive examination for

patients. On one hand, microscopic diagnosis requires experienced pathologists' exhaustive scrutiny of hematoxylin and eosin-stained (H&E) slides. Moreover, histological diagnosis of glioma is subjected to interobserver variation, and routine review of histological diagnosis by multiple pathologists is recommended[4,5]. On the other hand, molecular diagnosis necessitates invasive surgical resection/biopsy for glioma tissue followed by Sanger sequencing[6] and fluorescence in situ hybridization (FISH)[7], which are not always available in routine examinations of many medical centers.

The development of digitized scanners allows glass slides to be translated into whole-slide images (WSIs), which offers an opportunity for image analysis algorithms to achieve automatic and unbiased computational pathology. Most existing WSI-based diagnosis models adopt a deep-learning technique named convolutional neural network (CNN) for image recognition[8–10]. For glioma, several pathological CNN models have been proposed, such as a grading model trained on a small public dataset to distinguish glioblastoma and lower-grade glioma[11], a diagnostic platform developed on 323 patients to classify five subtypes according to the 2007 WHO criteria[12], a model trained on The Cancer Genome Atlas dataset to classify the three major types of glioma based on the 2021 WHO standard[13], and a histopathological auxiliary system for classification of brain tumors[14]. However, a WSI diagnostic model for detailed classification of adult-type diffuse glioma strictly according to the 2021 WHO rule is still in demand. Previous evidence has shown histopathological image features in glioma are associated with specific molecular alterations such as the IDH mutation[15–18]. However, as each genotype may share overlapping histological features on H&E sections (e.g., IDH-wildtype and IDH-mutant tumors), developing an integrated diagnosis model directly from WSI to classify the 2021 WHO types that combine both pathological and molecular features is still challenging.

Furthermore, there are unique challenges in CNN diagnosis using WSIs due to their gigapixel-level resolution, which makes original CNN computationally impossible. To tackle this obstacle, a WSI can be tiled into many small patches, from which a subset of cancerous patches can be selected from manually annotated pixel-level regions of interest (ROI). To avoid the heavy burden of manual annotation, weakly supervised learning techniques were applied to train WSI-CNNs with slide- or patch-level coarse labels such as cancer or non-cancer[10,18–25].

In this work, we propose a neuropathologist-level integrated diagnosis model for automatically predicting 2021 WHO types and grades of adult-type diffuse gliomas from annotation-free standard WSIs. The model avoids the annotation burden by using patient-level tumor types directly as weak supervision labels while exploiting the type-discriminative patterns by leveraging a feature domain clustering. The integrated diagnosis model is developed and externally tested using 2624 patients with adult-type diffuse gliomas from three hospitals. All datasets have integrated histopathological and molecular information strictly required for 2021 WHO classification. Our study provides an integrated diagnosis model for automated and unbiased classification of adult-type diffuse gliomas.

## Results
### Overview and patient characteristics
There were three datasets included in this study: Dataset 1 contained 1991 consecutive patients from the First Affiliated Hospital of Zhengzhou University (FAHZZU), Dataset 2 contained 305 consecutive patients from Henan Provincial People's Hospital (HPPH), and Dataset 3 contained 328 consecutive patients from Xuanwu Hospital Capital Medical University (XHCMU). The selection pipeline was shown in Fig. 1a. Therefore, a total of 2624 patients were included in this study as the study dataset (mean age, 50.97 years ± 13.04 [standard deviation]; 1511 male patients), including 503 A, 445 O, and 1676 GBM (Fig. 1b). The study dataset comprised a training cohort (n = 1362, mean age, 50.66 years ± 12.91; 787 men) from FAHZZU, a validation cohort (n = 340,

mean age, 50.81 years ± 12.33; 195 men) from FAHZZU, an internal testing cohort (n = 289, mean age, 50.25 years ± 13.08; 172 men) from FAHZZU, an external testing cohort 1 (n = 305, mean age, 52.46 years ± 12.82; 171 men) from HPPH, and external testing cohort 2 (n = 328, mean age, 50.82 years ± 14.25; 186 men) from XHCMU. The datasets were described in detail in Supplementary Methods A1. The clinical characteristics and integrated pathological diagnosis of the four cohorts are summarized in Supplementary Table 1. The detailed protocols for molecular testing are described in Supplementary Methods A2–A3. Representative results of IDH1/IDH2 mutations, 1p/19q deletions, CDKN2A homozygous deletion, EGFR amplification, and Chromosome 7 gain/Chromosome 10 loss are depicted in Supplementary Figs. 1–4. The integrated classification pipeline according to the 2021 WHO rule was shown in Fig. 2 and described in Supplementary Methods A4. There was no significant difference in type, grade, gender, age, and IDH mutation status among the training cohort, internal validation cohort, and internal testing cohort (two-sided Wilcoxon test or Chi-square test P-value > 0.05).

### Patch clustering-based integrated diagnosis model building
To select a subset of discriminative patches from a WSI, we clustered the patches based on their phenotypes and distinguished the more discriminative ones. The pipeline consisted of four steps: patch clustering, patch selection, patch-level classification, and patient-level classification, as shown in Fig. 1c. The clustering process can be found in Supplementary Methods A5. The CNN architecture and training parameters for patch selection were described in Supplementary Methods A6.

In the training cohort, 644,896 patches were extracted in total. Using a subset of 43653 patches from 100 randomly selected patients in the training cohort, a K-means clustering model was developed, where both the silhouette coefficient and the Calinski-Harabasz index reached their highest value at the optimal cluster number of nine, as shown in Fig. 3a, b. Using the K-mean algorithm, all 644,896 patches from the training cohort were partitioned into nine clusters. Correspondingly, nine separate patch-level CNN classifiers were obtained, and their patch-level accuracy in classifying the six categories was shown in Fig. 3c. Among them, three classifiers trained on cluster 2,5,7 had higher accuracy than the benchmark classifier (shown by the green bar in Fig. 3c). Therefore, the three clusters containing 275,741 patches in training cohort were selected for building the final patch-level classifier. The clustering results for three representative patients are shown in Fig. 3d. It showed the patch heterogeneity across clusters, implying the capability of the clustering-based method in distinguishing different image patterns. The tumor classification performance of the patch-level classifier built on the three selected clusters in each cohort is shown in Supplementary Fig. 5.

### Classification performance of the integrated diagnosis model
The diagnostic model was obtained by aggregating the patch-level classifications into patient-level results. We first showed the patient-level cross-validation results. The ROC curves for each fold and the mean ROC curves over all folds for classifying the six categories on the validation cohort were shown in Supplementary Fig. 6. The boxplots of AUCs in all folds were shown in Supplementary Fig. 7. The results demonstrated the model stability across different folds. Next, we assessed the performance of the best model (the fifth model, corresponding to ROC curves for fold 5 in Supplementary Fig. 6) selected in cross-validation on multiple testing cohorts. In classifying the six categories (task 1) of A Grade 2, A Grade 3, A Grade 4, O Grade 2, O Grade 3, and GBM Grade 4 (short for A2, A3, A4, O2, O3, and GBM), the model achieved corresponding AUCs of 0.959, 0.995, 0.953, 0.978, 0.982, 0.960 on internal validation cohort, 0.970, 0.973, 0.994, 0.932, 0.980, 0.980 on internal testing cohort, 0.934, 0.923, 0.987, 0.964, 0.978, 0.984 on external testing cohort 1, and 0.945, 0.944, 0.904,

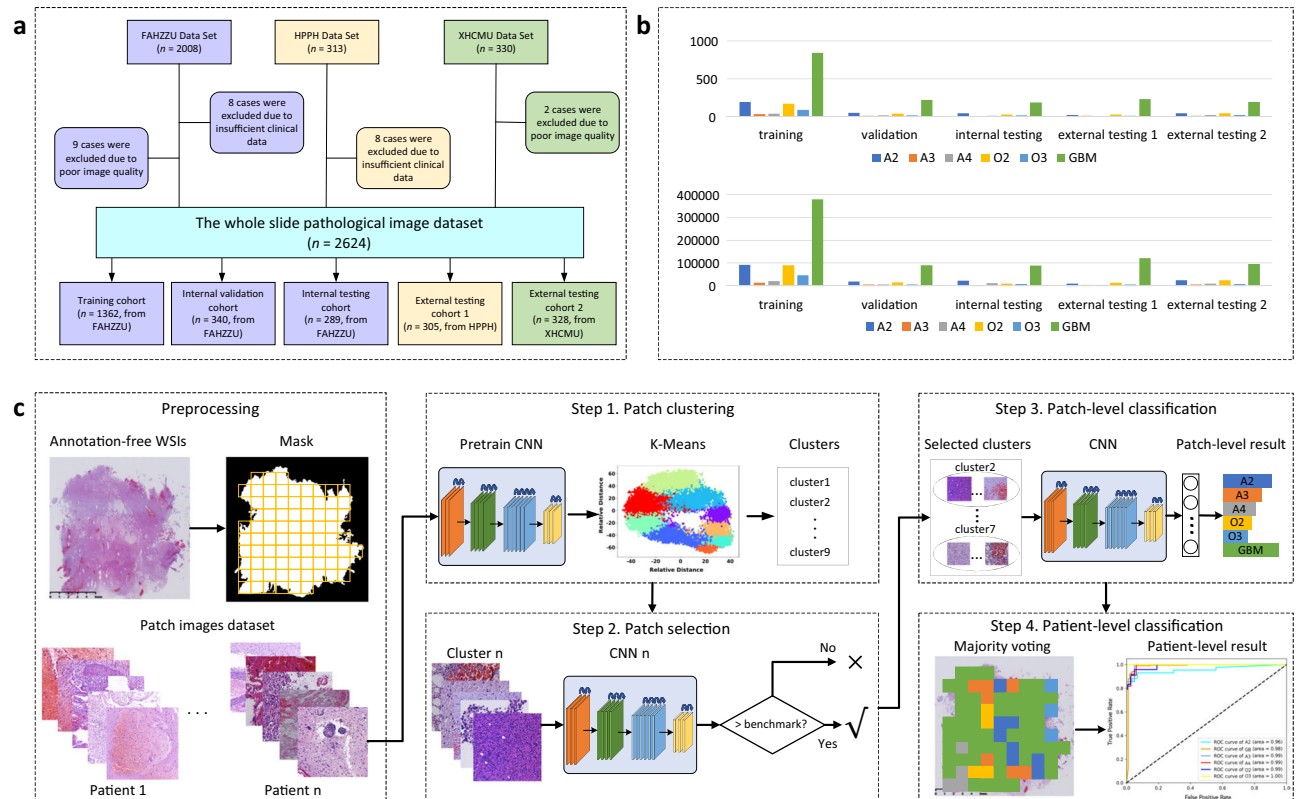

**Fig. 1 | Dataset and pipeline of the study. a** The patient selection procedure. **b** The bar graph of the patient/patch number for each tumor type in each cohort. **c** The pipeline of the presented clustering-based annotation-free classification method. FAHZZU First Affiliated Hospital of Zhengzhou University, HPPH Henan Provincial People's Hospital, XHCMU Xuanwu Hospital Capital Medical University. A2 Astrocytoma, IDH-mutant, Grade 2; A3 Astrocytoma, IDH-mutant, Grade 3; A4 Astrocytoma, IDH-mutant, Grade 4; O2 IDH-mutant, and 1p/19q-codeleted, Grade 2; O3 IDH-mutant, and 1p/19q-codeleted, Grade 3; GBM Glioblastoma, IDH-wildtype, Grade 4.

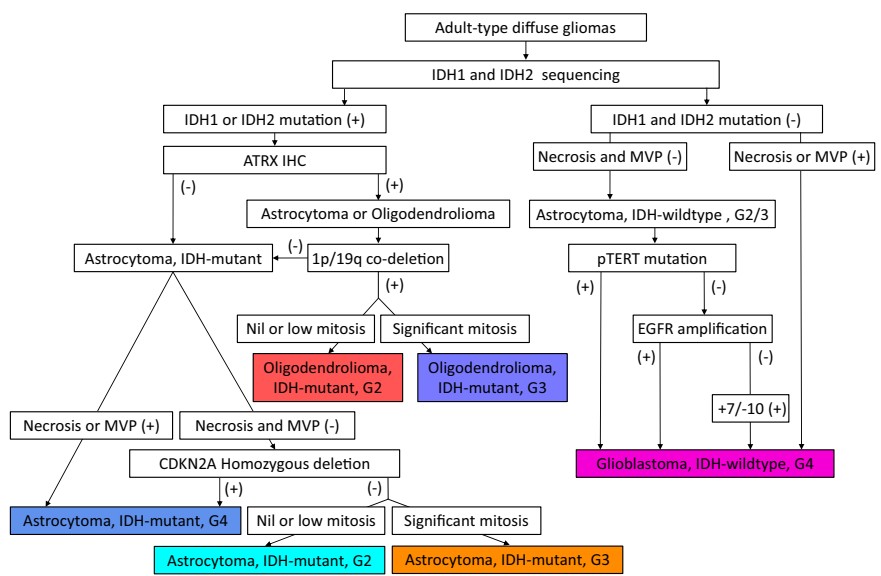

**Fig. 2 | Schematic showing the classification of adult-type diffuse gliomas in our study.** All tumors included in our study underwent the same classification pipeline. G2/3/4: grade 2/3/4; MVP: microvascular proliferation; +10/-7: whole chromosome 7 gain and whole chromosome 10 loss.

0.942, 0.950, 0.952 on external testing cohort 2, respectively, as shown in Fig. 4a−d and Table 1. In classifying the three types of A, O, and GBM while neglecting grades (task 2), the model achieved corresponding AUCs of 0.961, 0.974 and 0.960 on internal validation cohort, 0.969, 0.974, 0.980 on internal testing cohort, and 0.938, 0.973 and 0.983 on external testing cohort 1, and 0.941, 0.938 and 0.952 on external testing cohort 2, respectively, as shown in Fig. 4e−h and Table 1. The PR curves of the diagnostic model related to task 1 and task 2 were shown in Supplementary Fig. 8, demonstrating the model performance in this data imbalance problem.

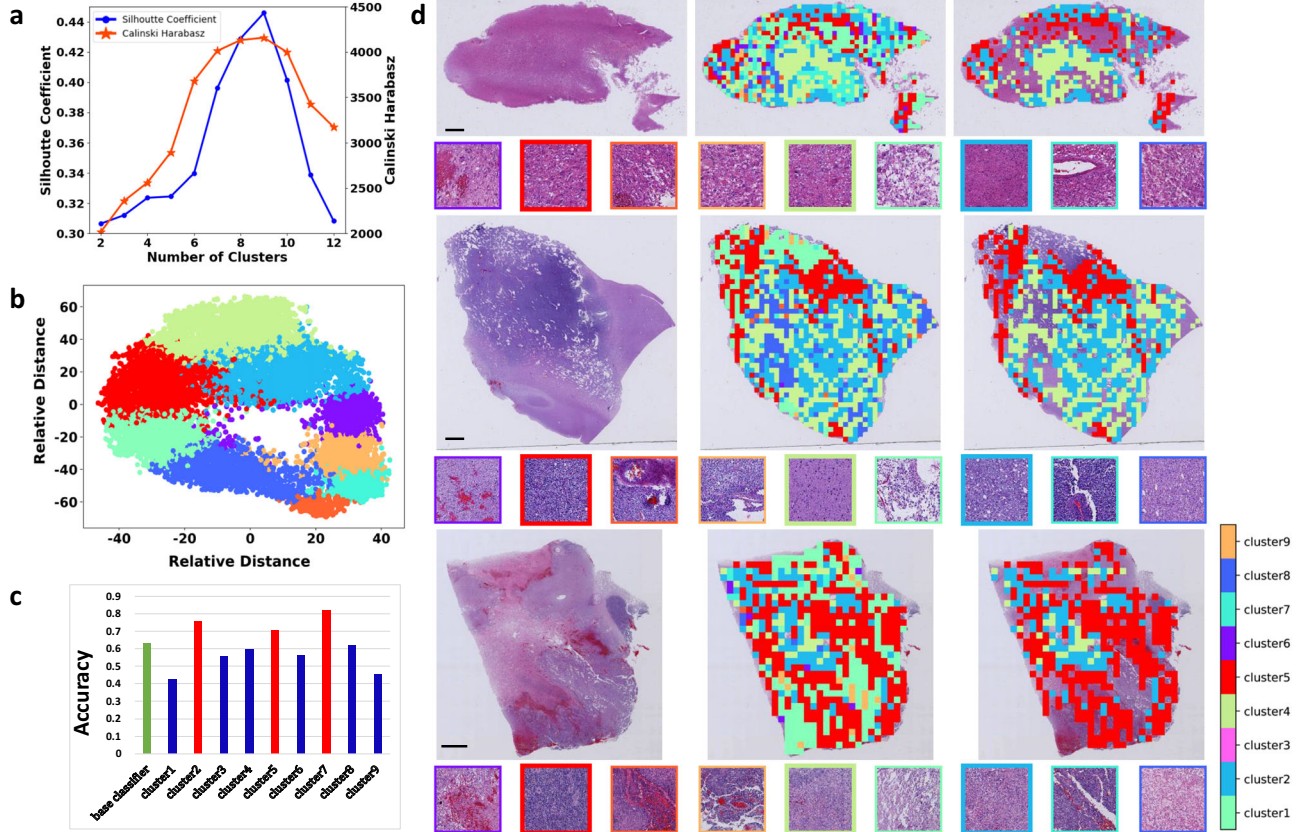

**Fig. 3 | The results of patch clustering and patch selecting. a** The silhouette plot (left) and the Calinski-Harabasz index plot (right) of the *K*-means clustering method with the cluster number ranging from 2 to 12. The silhouette coefficient, whose value ranges from -1 to 1, is used to assess the goodness of a clustering. A higher silhouette coefficient means better clustering. The Calinski-Harabasz index is calculated as the ratio of the between-cluster variance to the within-cluster variance. Similarly, a higher value of the Calinski-Harabasz index indicates better clustering performance. Well-grouped clusters are apart from each other and clearly distinguished. The silhouette coefficient and the Calinski-Harabasz index achieved their highest values of 0.447 and 4159.3, respectively, both at an optimal cluster number of nine. **b** Visualization of the nine clusters of the 43653 patches from 100 randomly selected patients in the training cohort. **c** Bar graph of patch-level classification accuracy of nine separate cluster-based classifiers. Three classifiers (shown by the red bar) trained on clusters 2,5,7 had higher accuracy than the benchmark classifier (shown by the green bar). Then, the patches within the three clusters 2,5,7 for each patient were selected for building the patient-level classifier. **d** The result of patch clustering and patch selection for three representative patients (top: A2; middle: O2; bottom: GBM). For each patient, the three images in the first row from left to right are the original whole-slide image, the distribution of the clustered patches (each color indicates a cluster), and the finally selected patches in the three clusters, respectively; the nine small images framed with different colors in the second row are representative patches from each of the nine clusters, where the finally selected three patches are shown in bold frames. Each experiment was repeated independently three times with the same results. Source data are provided as a Source Data file. Scale bars, 2 mm. The size of the patch is 1024 × 1024 pixels, with each pixel representing 0.50 microns.

Considering that IDH-wildtype diffuse astrocytic tumors without the histological features of glioblastoma but with TERT promoter mutations, EGFR amplification, or Chromosome 7 gain/Chromosome 10 loss (classified as glioblastomas in 2021 standard) may share similar histological features with the IDH-mutant Grade 2–3 astrocytoma, we also assessed the model's ability in classifying these two categories (task 3). In these two subgroups, our model achieved high performance with AUCs ranging from 0.935 to 0.984 in all cohorts, as shown in Fig. 4i–l and Table 1. On the other hand, the IDH-mutant glioblastoma in the 2016 WHO classification is classified as IDH-mutant astrocytoma grade 4 in the 2021 WHO classification, which may share similar histological features such as microvascular proliferation with IDH-wildtype glioblastoma. Our model also achieved good performance in distinguishing these two subgroups with AUCs ranging from 0.943 to 0.998 on all cohorts, as shown in Fig. 4m–p and Table 1 (task 4).

Furthermore, we assessed the model performance in classifying tumor grades within the type. In classifying A2, A3, and A4 within the IDH-mutant astrocytoma subgroup (task 5), the model achieved high AUCs ranging from 0.907 to 0.998 across all grades on all cohorts, as shown in Supplementary Fig. 9a–d and Table 1. In classifying O2 and O3 within the oligodendroglioma subgroup (task 6), the model maintained high AUCs ranging from 0.928 to 0.989 on all cohorts, as shown in Supplementary Fig. 9e–h and Table 1. Moreover, we also assessed the performance in distinguishing IDH-mutant diffuse astrocytoma with IDH-mutant 1p/19q-codeleted oligodendroglioma (task 7), achieving subgroup AUCs ranging from 0.957 to 0.994 on all cohorts, as shown in Supplementary Fig. 9i–l and Table 1.

**Comparison with other classification models**
The performance of the proposed clustering-based model was further compared with four previous models, a weakly supervised classical multiple-instance learning (MIL) model[8,9], an attention-based MIL (AMIL) model[26], a clustering-constrained-attention MIL (CLAM)[20], and the all-patch classification model. The AUCs of the classical MIL model and the all-patch model on all cohorts ranged from 0.793 to 0.997 in classifying the six categories (task 1) while ranged from 0.894 to 0.981 in classifying the three major types (task 2), as shown in Supplementary Figs. 10 and 11 and Supplementary Data 1 and 2. The two advanced methods, AMIL and CLAM, did not show significant improvement in AUCs in tasks 1 and 2 compared with classical MIL, as shown in Supplementary Figs. 12 and 13, respectively. The AUCs of all five models

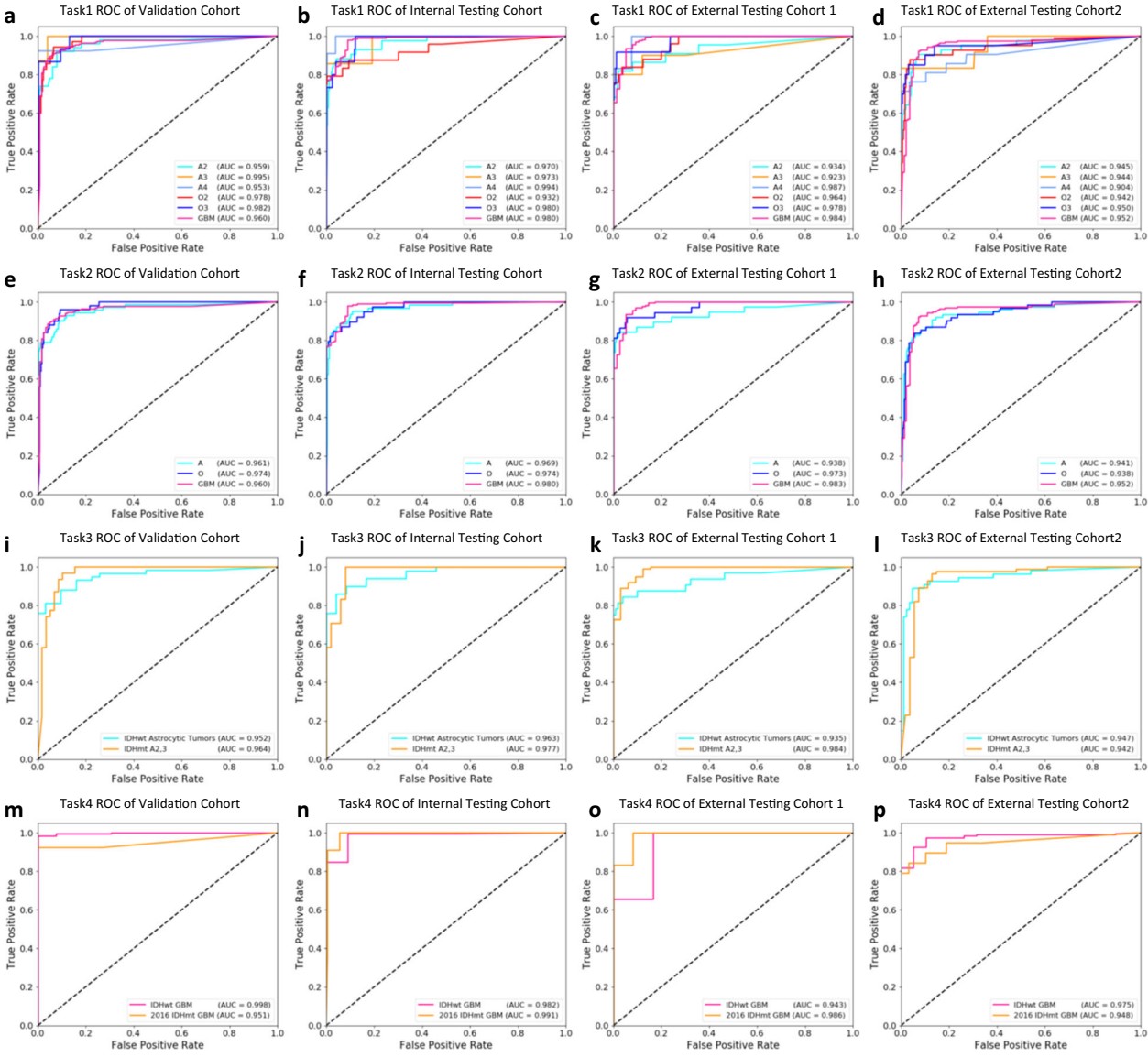

**Fig. 4 | The patient-level classification performance of the presented clustering-based diagnostic model on tasks 1-4.** (a–d, task 1): ROC curves for classifying the six categories of A2,3,4, O2,3, and GBM on the internal validation cohort (**a**), internal testing cohort (**b**), external testing cohort 1 (**c**), and external testing cohort 2 (**d**), respectively. (e–h, task 2): ROC curves for classifying the three major types of A, O, and GBM on the internal validation cohort (**e**), internal testing cohort (**f**), external testing cohort 1 (**g**), and external testing cohort 2 (**h**), respectively. (i–l, task 3): ROC curves for distinguishing the two subgroups of IDH-mutant astrocytic tumors A2-3 and IDH-wildtype diffuse astrocytic tumors without the histological features of glioblastoma (classified as glioblastoma) on the internal validation cohort (**i**), internal testing cohort (**j**), external testing cohort 1 (**k**), and external testing cohort 2 (**l**), respectively. (**m**–**p**, task 4): ROC curves for distinguishing the two subgroups of IDH-mutant astrocytic tumors A2 and IDH-wildtype gliomas on the internal validation cohort (**m**), internal testing cohort (**n**), external testing cohort 1 (**o**), and external testing cohort 2 (**p**), respectively. Corresponding classification results can be found in Table 1. O2 IDH-mutant, and 1p/19q-codeleted, Grade 2; A2 Astrocytoma, IDH-mutant, Grade 2; O3 IDH-mutant, and 1p/19q-codeleted, Grade 3; A3 Astrocytoma, IDH-mutant, Grade 3; A4 Astrocytoma, IDH-mutant, Grade 4; GBM glioblastoma, IDH-wildtype, Grade 4. Source data are provided as a Source Data file.

were summarized in Supplementary Table 2. The results of the Delong analysis between the AUCs of the clustering-based model and other models were summarized in Supplementary Table 3. In classifying tumor grades within types, distinguishing IDH-mutant astrocytoma with IDH-mutant 1p/19q-codeleted oligodendroglioma, and distinguishing IDH-mutant astrocytoma with astrocytoma-like IDH-wildtype glioblastoma, the performance of the MIL model and the all-patch model was summarized in Supplementary Figs. 14 and 15 and Supplementary Data 1 and 2 (tasks 3–7). Among the five models, the MIL model and its two variants were numerically inferior to or comparable with the clustering-based model, while the all-patch model lagged the other four models in all tasks. As shown in Supplementary Table 3, on most datasets the difference in AUCs between the clustering-based

model and each of the three MIL models was not significant (Delong $P > 0.05$) in classifying the six tumor types (task 1). In classifying the three types (task 2), the AUC of the clustering-based model was significantly higher than that of the all-patch model on all testing datasets (Delong $P < 0.05$).

## Interpretation of the CNN classification
To visualize and interpret the relative importance of different regions in classifying the tumors, the class activation maps (CAM) along with the corresponding patches and WSIs from ten representative patients were shown in Fig. 5. The CAM highlighted in red which regions contributed most to the classification task. These highlighted regions were then evaluated and interpreted from neuropathologist's perspectives.

**Table 1 | The classification performance of the proposed integrated diagnostic model**

| Task | Types/grades | AUC | Accuracy | Sensitivity | Specificity | F1-score |
|---|---|---|---|---|---|---|
| 1 | A2 | 0.959 | 0.717 | 0.760 | 0.948 | 0.738 |
| | | 0.970 | 0.814 | 0.814 | 0.967 | 0.814 |
| | | 0.934 | 0.850 | 0.773 | 0.989 | 0.810 |
| | | 0.945 | 0.756 | 0.738 | 0.965 | 0.747 |
| | A3 | 0.995 | 1.000 | 0.875 | 1.000 | 0.933 |
| | | 0.973 | 0.857 | 0.857 | 0.996 | 0.857 |
| | | 0.923 | 1.000 | 0.800 | 1.000 | 0.889 |
| | | 0.944 | 0.833 | 0.833 | 0.994 | 0.833 |
| | A4 | 0.953 | 0.857 | 0.923 | 0.944 | 0.889 |
| | | 0.994 | 0.714 | 0.909 | 0.986 | 0.800 |
| | | 0.987 | 1.000 | 0.833 | 1.000 | 0.909 |
| | | 0.904 | 0.882 | 0.714 | 0.993 | 0.789 |
| | O2 | 0.978 | 0.682 | 0.857 | 0.954 | 0.760 |
| | | 0.932 | 0.760 | 0.792 | 0.977 | 0.776 |
| | | 0.965 | 0.950 | 0.760 | 0.996 | 0.844 |
| | | 0.942 | 0.810 | 0.829 | 0.972 | 0.819 |
| | O3 | 0.982 | 0.812 | 0.867 | 0.991 | 0.839 |
| | | 0.980 | 0.579 | 0.733 | 0.971 | 0.647 |
| | | 0.978 | 1.000 | 0.750 | 1.000 | 0.857 |
| | | 0.950 | 0.778 | 0.700 | 0.987 | 0.737 |
| | GBM | 0.960 | 0.956 | 0.900 | 0.926 | 0.927 |
| | | 0.980 | 0.956 | 0.915 | 0.920 | 0.935 |
| | | 0.983 | 0.947 | 1.000 | 0.827 | 0.973 |
| | | 0.952 | 0.914 | 0.943 | 0.875 | 0.928 |
| 2 | A | 0.961 | 0.770 | 0.803 | 0.937 | 0.786 |
| | | 0.969 | 0.812 | 0.852 | 0.947 | 0.832 |
| | | 0.938 | 0.939 | 0.816 | 0.993 | 0.873 |
| | | 0.941 | 0.843 | 0.787 | 0.957 | 0.814 |
| | O | 0.974 | 0.733 | 0.880 | 0.945 | 0.800 |
| | | 0.974 | 0.727 | 0.821 | 0.952 | 0.771 |
| | | 0.973 | 1.000 | 0.784 | 1.000 | 0.879 |
| | | 0.938 | 0.800 | 0.787 | 0.955 | 0.793 |
| | GBM | 0.960 | 0.956 | 0.900 | 0.926 | 0.927 |
| | | 0.980 | 0.956 | 0.915 | 0.920 | 0.935 |
| | | 0.983 | 0.947 | 1.000 | 0.827 | 0.973 |
| | | 0.952 | 0.914 | 0.943 | 0.875 | 0.928 |
| 3 | IDHwt Astrocytic Tumors | 0.952 | 0.963 | 0.897 | 0.935 | 0.929 |
| | | 0.963 | 0.976 | 0.820 | 0.958 | 0.891 |
| | | 0.935 | 1.000 | 0.812 | 1.000 | 0.896 |
| | | 0.947 | 0.956 | 0.796 | 0.976 | 0.869 |
| | IDHmt A2,3 (A4 excluded) | 0.964 | 0.829 | 0.935 | 0.897 | 0.879 |
| | | 0.977 | 0.719 | 0.958 | 0.820 | 0.821 |
| | | 0.984 | 0.943 | 1.000 | 0.812 | 0.971 |
| | | 0.942 | 0.880 | 0.976 | 0.796 | 0.926 |
| 4 | IDHwt GBM | 0.998 | 0.995 | 0.995 | 0.923 | 0.995 |
| | | 0.982 | 0.994 | 0.983 | 0.909 | 0.988 |
| | | 0.943 | 0.996 | 1.000 | 0.833 | 0.998 |
| | | 0.975 | 0.979 | 1.000 | 0.789 | 0.989 |
| | 2016 IDHmt GBM (A4 in 2021 rule) | 0.951 | 0.923 | 0.923 | 0.995 | 0.923 |
| | | 0.991 | 0.769 | 0.909 | 0.983 | 0.833 |
| | | 0.986 | 1.000 | 0.833 | 1.000 | 0.909 |
| | | 0.948 | 1.000 | 0.789 | 1.000 | 0.882 |
| 5 | A2 | 0.979 | 1.000 | 0.760 | 1.000 | 0.864 |
| | | 0.990 | 1.000 | 0.814 | 1.000 | 0.897 |
| | | 0.939 | 0.944 | 0.773 | 0.938 | 0.850 |
| | | 0.929 | 0.939 | 0.738 | 0.939 | 0.826 |
| | A3 | 0.994 | 1.000 | 0.875 | 1.000 | 0.933 |
| | | 0.981 | 1.000 | 0.857 | 1.000 | 0.923 |
| | | 0.930 | 1.000 | 0.800 | 1.000 | 0.889 |
| | | 0.921 | 0.524 | 0.917 | 0.841 | 0.667 |
| | A4 | 0.958 | 0.500 | 1.000 | 0.776 | 0.667 |
| | | 0.998 | 0.550 | 1.000 | 0.820 | 0.710 |
| | | 0.990 | 0.500 | 1.000 | 0.812 | 0.667 |
| | | 0.907 | 0.762 | 0.762 | 0.907 | 0.762 |
| 6 | O2 | 0.987 | 0.968 | 0.857 | 0.933 | 0.909 |
| | | 0.928 | 1.000 | 0.792 | 1.000 | 0.884 |
| | | 0.947 | 0.950 | 0.760 | 0.917 | 0.844 |
| | | 0.948 | 0.895 | 0.850 | 0.800 | 0.872 |
| | O3 | 0.989 | 0.737 | 0.933 | 0.857 | 0.823 |
| | | 0.964 | 0.750 | 1.000 | 0.792 | 0.857 |
| | | 0.967 | 0.647 | 0.917 | 0.760 | 0.759 |
| | | 0.956 | 0.727 | 0.800 | 0.850 | 0.762 |
| 7 | A | 0.982 | 0.919 | 0.905 | 0.898 | 0.912 |
| | | 0.983 | 0.945 | 0.912 | 0.914 | 0.928 |
| | | 0.994 | 0.939 | 1.000 | 0.935 | 0.969 |
| | | 0.957 | 0.913 | 0.926 | 0.893 | 0.919 |
| | O | 0.978 | 0.880 | 0.898 | 0.905 | 0.889 |
| | | 0.986 | 0.865 | 0.914 | 0.912 | 0.889 |
| | | 0.990 | 1.000 | 0.935 | 1.000 | 0.966 |
| | | 0.960 | 0.909 | 0.893 | 0.926 | 0.901 |

The top, second, third, and bottom rows for each type/grade indicate the performance of the internal validation cohort, internal testing cohort, and external testing cohorts 1 and 2, respectively. Task 1: classifying the six categories. Task 2: classifying the three types. Task 3: classifying IDH-wildtype diffuse astrocytic tumors and IDH-mutant astrocytoma grade 2 and 3. Task 4: classifying IDH-mutant GBM in 2016 WHO classification (classified as IDH-mutant A4 in 2021 rule) and IDH-wildtype GBM. Task 5–6: classifying grades within types. Task 7: classifying IDH-mutant A and IDH-mutant 1p/19q-codeleted O.

As shown in Fig. 5, the ten examples were assigned to five groups, where the two examples in each group shared the same grades or histological features. This human-readable CAM indicated that the classification basis of the clustering-based model generally aligned with pathological morphology well recognized by pathologists. For example, in distinguishing O2 from A2 or O3 from A3, our model generally highlighted morphological characteristics of oligodendrocytes/astrocytes, which were consistent with human expertize. We also observed that in classifying cases with shared histological features including necrosis and microvascular proliferation, features that might reflect underlying IDH mutations and CDKN2A homozygous deletion can be captured by our model. These features may offer potential predictive value and might be useful in assisting human readers in achieving more accurate diagnoses.

## Discussion

In this study, we presented a CNN-based integrated diagnosis model that was capable of automatically classifying adult-type diffuse gliomas according to the 2021 WHO standard from annotated-free WSIs. We compiled a large dataset including 2624 patients with both histological and molecular information. Extensive validation and comparative studies confirmed the accuracy and generalization ability of our model.

Compared to previous work, our research had several strengths by addressing the key challenges in computational pathology: (1) The deep-learning model can be trained with only tumor types as weakly supervised labels by using a patch clustering technique, which obviated the burden of pixel-level or patch-level annotations. (2) Using only pathological images, our model enables high-performance integrated diagnosis that traditionally requires combining pathological and molecular information. This was made possible through a clustering-based CNN that can learn imaging features containing both pathological morphology and underlying biological clues. (3) Using a large training dataset including 644896 patch images from 1362 patients, our model can generalize to an internal testing cohort and two external testing cohorts, with strong performance in classifying major types, grades within type, and especially in distinguishing genotypes with shared histological features.

Several WSI CNN models have been developed for predicting histological grades according to the 2007 WHO classification in patients with glioma[11,12,27,28]. For instance, Ertosun et al. applied CNN to perform binary classification between glioblastoma and lower-grade glioma with an accuracy of 96%, and between grade II and III glioma with an accuracy of 71%[11]. Jin et al. presented a diagnostic platform to classify five major categories considering both histological grades and molecular makers based on 323 patients, with an accuracy of 87.5%[12]. However, to date, there are no CNN-based integrated diagnostic models strictly according to the 2021 WHO classification, which introduces substantial changes compared to previous editions. Jose L et al. developed a CNN model using The Cancer Genome Atlas dataset to classify three types of gliomas considering two molecular markers (IDH mutation and 1p/19q codeletion) based on the 2021 WHO standard, with an accuracy of 86.1% and an AUC of 0.961[13]. Our CNN model is the one that can classify gliomas into six types strictly adhering to the 2021 rule. To achieve this, we collected a much larger dataset and performed the integrated diagnosis for each patient according to the 2021 WHO criteria, where more comprehensive molecular information including IDH mutation, 1p/19q codeletion, CDKN2A homozygous deletion, TERT promoter mutation, EGFR amplification, and Chromosome 7 gain/Chromosome 10 loss were obtained to determine the types.

To emphasize the integrated diagnosis, the 2021 edition introduces a new "grades within type" classification system, where both grades and types are determined by combining histological and molecular information. In our study, we predicted the tumor grades/

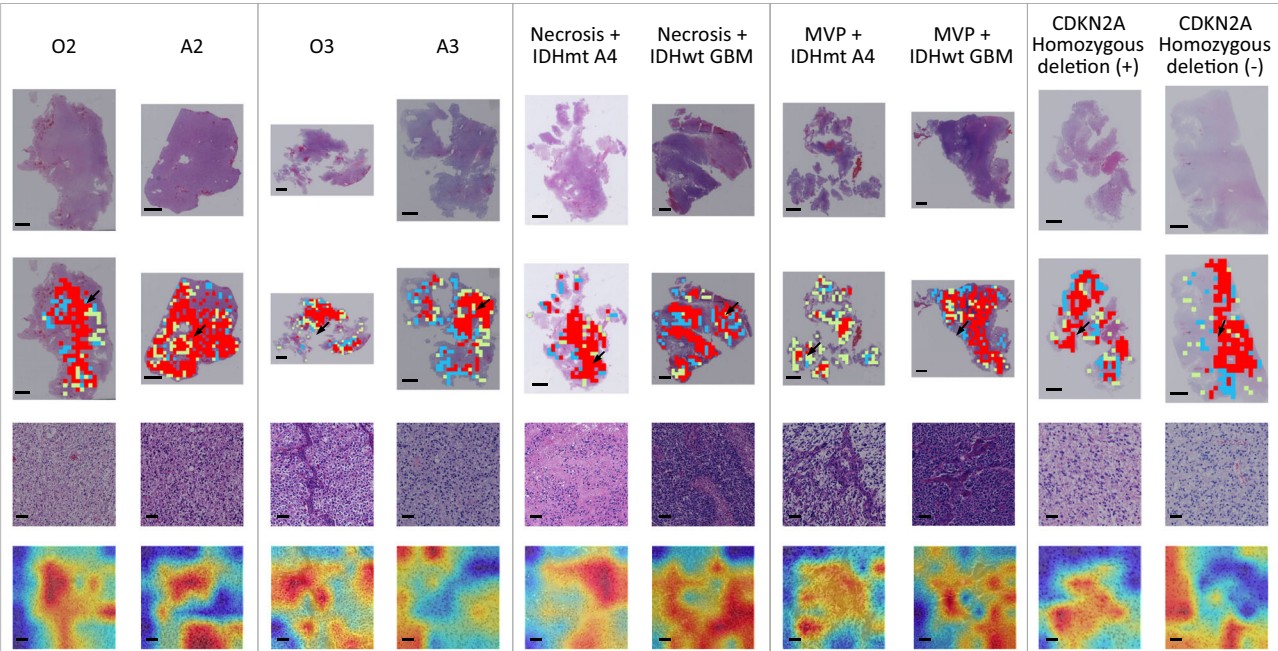

**Fig. 5 | Ten examples of classification results on the external testing sets.** The ten examples were assigned into five groups, where the two examples in each group shared the grades or histological features. The first row represents the original whole-slide images. The second row shows selected patches from the three clusters used for building the diagnostic model, where each color indicates a cluster. The third row shows a representative patch indicated by an arrow in the second row. The fourth row shows the class activation maps (CAM) generated by the diagnostic model overlapped on their corresponding patches. Regions with warm colors refer to the areas on which our model is focused on the typical area we are interested in for each group. O2 IDH-mutant, and 1p/19q-codeleted, Grade 2; A2 Astrocytoma, IDH-mutant, Grade 2; O3 IDH-mutant, and 1p/19q-codeleted, Grade 3; A3 Astrocytoma, IDH-mutant, Grade 3; A4 Astrocytoma, IDH-mutant, Grade 4; GBM: Glioblastoma, IDH-wildtype, Grade 4; IDHmt: IDH mutation; IDHwt IDH wildtype; MVP, microvascular proliferation; (+): positive; (−): negative. Scale bars in subfigures in the first and second rows, 2 mm. Scale bars in subfigures in the third and fourth rows, 40μm. The size of the patch is 1024 × 1024 pixels, with each pixel representing 0.50 microns.

types directly from pathological images, and no molecular information was fed into the model. This implies that our model can learn molecular characteristics from pathological images to achieve an integrated diagnosis. Several studies have also shown the ability of CNN to recognize the genetic alterations directly from WSI, such as mutation detection[8,13–15,18,19], microsatellite instability prediction[29], and pancancer genetic profiling[30,31]. In a recent study on CNN-based pathological diagnosis[12], the glioma classification was extended from three histological grades to five categories by adding the IDH and 1p/19q status. However, it is not strict WHO-consistent integrated classification, and the dataset with molecular information is relatively small ($n = 296$). Generally, these studies indicated a potential link between the tumor's histopathological morphology and underlying molecular composition.

Our clustering-based CNN model dedicated to learning the most representative features from the entire WSI had two major advantages. First, it avoided the need for any manual annotation by automatically selecting several type-relevant patch clusters that contributed more to the integrated classification task. Second, it aggregated local features to reach a global diagnosis by selectively fusing the most discriminative information from multiple relevant patches. Traditionally, manual annotation is required to delineate cancerous regions of interest for CNN training[10]. However, the manual delineation is always time-consuming and subjective. To avoid pixel-level annotation, weakly supervised methods were developed where experts can assign a label to an image. Among them, MIL and its variants employing a "bag learning" strategy have been widely used in WSI classification[8,9]. Our study compared the presented clustering approach with the classical MIL and its two variants, the AMIL[26] and CLAM[20], demonstrating the superior performance of our approach in classifying the six integrated types, the three histological categories, and the grades within each

type. Especially, our clustering model also achieved high performance in classifying several histologically similar subgroups, i.e., IDH-mutated vs. IDH wildtype tumors with similar morphology, and 1p/19q codeleted vs. 1p/19q non-codeleted tumors with similar morphology. These new classifications are also the major changes introduced by the 2021 WHO rule. Furthermore, the attention mechanism incorporated in both AMIL and CLAM did not seem to bring as much benefit as expected. One reason might be the high degree of variability and complexity within the pathologic data, making it hard to learn effective attention weights for instances related to the target classes. Specifically, to classify the six types according to the 2021 WHO rule, the model needs to identify discriminative morphology related to histologic types (A, O, and GBM) and grades within types (A2/3/4, O2/3), and tumor genotypes with shared histologic features (e.g., IDH-wildtype and -mutant tumors). Furthermore, some key instances might be sparse (microvascular proliferation or necrosis). The discriminative features might be contained in the same instances, in many different instances, or in sparse instances. These key instances may be too diverse and complex to be recognized by an attention mechanism. Moreover, we guess that the label noise induced by the simplified slide-to-patch label assignment would also impair the attention weights to some extent. Instead of emphasizing key patches, we turned to searching for important patch clusters with similar imaging phenotypes. Our data as well as the CAM visualization suggested the capability of the clustering-based model in recognizing not only pathological morphology such as microvascular proliferation and necrosis useful for histological classification, but also imaging patterns reflecting underlying genomic alterations useful for the integrated diagnosis.

Despite the encouraging results, three limitations should be pointed out. First, despite our dataset comprises of a sample size

of 2624 patients from three hospitals, future international multi-center and multiracial dataset of a larger sample size is welcomed. Second, in our study, all slides from three hospitals were scanned using the same digital scanner to ensure consistency. To address the impact of scanner variability and develop a classifier with good robustness in clinical practice, we plan to collect a larger dataset of WSIs obtained from a variety of scanners. Advanced stain normalization may be required to enhance the model's robustness. We will also assess the impact of different stain normalization methods, as the variations in stain intensity may affect the performance of deep-learning models. Third, more preclinical experimental work in genome, transcriptome, proteome, and animal level is needed to further elucidate the biological interpretability of the deep-learning model.

In conclusion, our data suggested that the presented CNN model can achieve high-performance fully automated integrated diagnosis that adheres to the 2021 WHO classification from annotation-free WSI. Our model has the potential to be used in clinical scenarios for unbiased classification of adult-type diffuse gliomas.

## Methods

### Patients and datasets

This study was a part of the registered clinical trial (ClinicalTrials ID: NCT04217044). This study was approved by the Human Scientific Ethics Committee of the First Affiliated Hospital of Zhengzhou University (FAHZZU), Henan Provincial People's Hospital (HPPH), and Xuanwu Hospital Capital Medical University (XHCMU). Informed consent and participant compensation were waived by the Committee due to the retrospective and anonymous analysis. There were three datasets included in this study: Dataset 1 contained 1991 consecutive patients from FAHZZU, Dataset 2 contained 305 consecutive patients from HPPH, and Dataset 3 contained 328 consecutive patients from XHCMU. Dataset 1 includes three cohorts: a (1) training cohort ($n = 1362$, from FAHZZU) used to develop the glioma type/grade classification model, a (2) validation cohort ($n = 340$, from FAHZZU) used to optimize the model, and a (3) internal testing cohort ($n = 289$, form FAHZZU) used to test the model. The training and validation cohorts were selected with stratified random sampling from the FAHZZU patient set collected from January 2011 to December 2019 at a ratio of 4:1, where the clinical parameters between both cohorts were balanced. We repeated this procedure in a five-fold cross-validation approach, re-assigning the patients into training and validation cohorts five times. Patients from FAHZZU between January 2020 and December 2020 were used as the internal testing cohort. Dataset 2 was used as an external testing cohort 1, and dataset 3 was used as an external testing cohort 2. The datasets were described in detail in Supplementary Methods A1. The inclusion criteria are as follows: (1) adult patients (>18 years) surgically treated and pathologically diagnosed as diffuse gliomas (WHO Grade 2–4), (2) availability of clinical, histological, and molecular data, (3) availability of sufficient formalin-fixed, paraffin-embedded (FFPE) tumor tissues for testing for molecular markers in the 2021 WHO classification of adult-type diffuse gliomas, (4) availability of H&E slides for scanning as digitalized WSIs, (4) sufficient image quality of digitalized WSIs. The selection pipeline is shown in Fig. 1a.

### Determination of WHO classification

In the last 5 years since the publication of the 2016 Edition of the WHO CNS, the development of targeted sequencing and omics techniques has helped neuro-oncologists gradually establish some new tumor types in clinical practice, as well as a series of molecular markers. Based on 7 updates at the Consortium to Inform Molecular and Practical Approaches to CNS Tumor Taxonomy (cIMPACT-NOW), the International Agency for Research on Cancer (IARC) has finally released the 5th edition of the WHO Classification of Tumors of the CNS.

According to cIMPACT-NOW update 3[32], despite appearing histologically as grade II and III, IDH-wildtype diffuse astrocytic gliomas that contain high-level EGFR amplification (excluding low-level EGFR copy number gains, e.g., trisomy 7), or whole chromosome 7 gain and whole chromosome 10 loss (+7/−10), or TERT promoter mutations, correspond to WHO grade IV and should be referred to as diffuse astrocytic glioma, IDH-wildtype, with molecular features of glioblastoma, WHO grade 4. According to cIMPACT-NOW update 5[33], diffusely infiltrative astrocytic glioma with an IDH1 or IDH2 mutation that exhibits microvascular proliferation or necrosis or CDKN2A/B homozygous deletion or any combination of these features should be referred to as Astrocytoma, IDH-mutant, WHO grade 4. Thus, in 5th edition of the WHO CNS, adult-type diffuse gliomas are divided into (1) Astrocytoma, IDH-mutant, Grade 2,3,4; (2) Oligodendroglioma, IDH-mutant, and 1p/19q-codeleted, Grade 2,3 and (3) Glioblastoma, IDH-wildtype, Grade 4 (A2, A3, A4, O2, O3, and GBM)[2].

Therefore, in our study, formalin-fixed, paraffin-embedded (FFPE) tissues were used for the detection of ATRX by immunohistochemistry (IHC), and for detection of mutational hotspots in IDH1/IDH2 and TERT promoter by Sanger sequencing, as well as for detection of Chromosome 1p/19q, CDKN2A, EGFR and chromosome 7/10 status by fluorescence in situ hybridization (FISH). The detailed protocols are described in Supplementary Methods A2 and A3. The integrated classification pipeline according to the 2021 WHO rule is shown in Fig. 2 and described in Supplementary Methods A4.

### WSI data acquisition and preprocessing

The slides were scanned using the MAGSCAN-NER scanner (KF-PRO-005, KFBIO) to obtain the WSI. In our study, one patient had one WSI. As tissues generally occupy a portion of the slide with large areas of white background space in a WSI, tissue segmentation should be performed first. The WSI at the 5× resolution was transformed from RGB to Lab color space and the tissue was segmented with a threshold value calculated using the OSTU algorithm. The segmented tissue image was divided into many 1024 × 1024 patches at 20 × objective magnifications (0.5 microns per pixel). The patches were adjacent to one another covering the entire WSI. From all 2624 patients, a total of 1292420 patches were extracted, as shown in Fig. 1b. The number of patches in different WSIs varied from hundreds to more than 2000. Each WSI belonged to one of the six categories: A2, A3, A4, O2, O3, and GBM. This patient-level label was also assigned to each patch within one WSI. All classifiers in the following were trained to predict the six tumor types.

### Integrated diagnosis model building

We aimed to find a subset of discriminative patches from a WSI. Considering that a group of patches may share similar imaging patterns or phenotypes, we clustered the patches based on their phenotypes and distinguished the clusters with better discriminative power. The pipeline consisted of four steps: patch clustering, patch selection, patch-level classification, and patient-level classification, as shown in Fig. 1c.

**Patch clustering.** First, the patch clustering algorithm was trained using 43653 candidate patches from 100 randomly selected patients in the training cohort, including 11 A2, 2 A3, 2 A4, 14 O2, 3 O3, and 68 GBM patients. Considering that the original image may not present type-relevant cancer phenotypes, we chose to cluster the patches in the feature domain. The patches were resized into 256 × 256 and were fed into a pre-trained CNN for deep feature extraction. Here a ResNet-50 trained with patch-level labels (six categories) on all patches in the training cohort was used as the CNN feature extractor (referred to as all-patch classifier). Using this trained ResNet-50, 2048 deep features can be extracted from the averaging pooling layer for each patch. Based on the features, the candidate 43,653 patches for the 100

patients were used to develop a $K$-means clustering algorithm by partitioning these patches into $K$ clusters, where the optimal cluster number $K$ was determined using the silhouette coefficient. The Calinski-Harabasz index was also used to additionally assess the clustering quality. The patches in different clusters were considered to have discriminative imaging patterns related to cancer types. The clustering process can be found in Supplementary Methods A5.

**Patch selection.** Using the established $K$-means clustering algorithm, all patches from each patient in the training cohort were partitioned into $K$ clusters. Next, $K$ separate patch-level CNN classifiers were trained on the $K$ patch clusters for all patients in the training cohort respectively, where the ResNet-50 was used as the CNN architecture and the training parameters were the same as used in the all-patch classifier. The $K$ clusters obtained in the validation cohort were used to optimize the $K$ corresponding classifiers. The $K$ cluster-based classifiers may have different powers in classifying the tumor types. Here we used the all-patch classifier as a performance benchmark. For each patient, the clusters with better classification accuracy than the benchmark were selected for further analysis. The CNN architecture and training parameters were detailed in Supplementary Methods A6.

**Patch-level classification.** Using the patches from all selected clusters, a patch-level ResNet-50 model was trained on the training cohort while optimized on the validation cohort. The same training parameters were used. This network was used to provide an estimation of the tumor types for each input image patch. Next, we should aggregate the patch-level estimations to make a final patient-level prediction.

**Patient-level classification.** The patch-level predictions were aggregated to determine the types of the entire WSI using a majority voting approach. Specifically, the class to which the maximum number of patches belonged was used as the final patient-level prediction. This aggregation approach can reduce the bias of patch-level prediction.

**Model selection.** To assess the model's robustness and to select an optimal model, we repeated the training/validation cohort division procedure five times using five-fold cross-validation. In each repetition, the training and validation sets were divided using stratified random resampling with patient characteristics balanced between both sets. During the cross-validation process, the model was trained for a minimum of 50 epochs. Then, the loss on the validation set was computed in each epoch, where the model with the lowest average validation loss over 10 consecutive epochs was saved. If such a model was not found, the training continued up to a maximum of 150 epochs. Finally, the patient-level model with the best-averaging performance across all folds was selected as the proposed diagnostic model.

**Statistical analysis**
Statistical analysis was performed using Python (Version 3.6.1). $P$-value < 0.05 was considered significant. All data analysis was performed using Python 3.6.1. Specifically, the packages or software comprised PyTorch 1.10.0 for model training and testing, CUDA 11.6 and cuDNN 8.1.0.77 for GPU acceleration, and scikit-learn 1.0.2 for statistical analysis. All CNNs were trained on two NVIDIA Tesla V100 GPUs. The difference in patient characteristics between training and the other cohorts was assessed by a two-sided Wilcoxon test or Chi-square test. The patch-level classifiers were trained on the training cohort and optimized on the validation cohort. The performance of the optimal patient-level classifiers in five-fold cross-validation was further tested on the internal testing cohort and two external testing cohorts. Receiver operating characteristic (ROC) analysis was used for performance evaluation in terms of area under the ROC curve (AUC), accuracy, sensitivity, specificity, and F1-score in classifying the six categories A2, A3, A4, O2, O3, and GBM. These metrics were calculated

using a one-vs.-rest approach in the multi-class problem. The average AUC over the six categories on the validation cohort for each fold was used to select the best model in cross-validation. To address the class imbalance problem, the precision-recall (PR) curves were also calculated to comprehensively assess the model performance. In addition, the performance of the clustering-based model was compared with another four models, a weakly supervised classical multiple-instance learning (MIL) model[8,9], an attention-based MIL (AMIL) model[26], a clustering-constrained-attention MIL (CLAM)[20], and the all-patch classification model. Briefly, in MIL the patches with the highest score (that were most likely to be cancerous) were selected for diagnosis model building. AMIL and CLAM were two variants of MIL, where the former learned to emphasize the patches related to the target classes while the latter extended AMIL to a general multi-class with a refined feature space. As described before, the all-patch model used all patches for classification without patch selection. The statistical difference between AUCs was compared using DeLong analysis. Reporting of the study adhered to the STARD guideline[34].

### Reporting summary
Further information on research design is available in the Nature Portfolio Reporting Summary linked to this article.

## Data availability
The whole-slide histology image data and paired pathological data from First Affiliated Hospital of Zhengzhou University, Henan Provincial People's Hospital, and Xuanwu Hospital Capital Medical University are protected and restricted to be used with institutional permission and are therefore not publicly available due to data privacy policies. For example, whole-slide histology image data of six representative patients used for model testing was uploaded with the code and is publicly available in CodeOcean database[35]. Sanger sequencing data for IDH and TERT promoter mutations have been deposited in the Genome Sequence Archive (GSA for Human) database under ID HRA005239. Z.-Y.Z. or W.-C.L. should be contacted to request access to the WSI data and Sanger sequencing raw data. Requests will be assessed according to institutional policies to determine whether the data request is subject to patient privacy obligations. A user agreement will be required. All other relevant data supporting the key findings of this study are available within the article and its Supplementary Information files or from the corresponding authors upon request. Source data are provided in this paper.

## Code availability
The Python codes for implementation and testing of the model used to calculate the classification probability as well as the testing data were deposited into a publicly available repository at https://doi.org/10.24433/CO.1134119.v1[35].

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

## Acknowledgements

This research was supported by the Key-Area Research and Develop-ment Program of Guangdong Province 2021B0101420006 (Z.-C.L.), the National Natural Science Foundation of China U20A2017 (Z.-C. L.), 82273493 (Z.-Y.Z.), and U1904148 (W.-W.W.), the Natural Science Foundation of Henan Province for Excellent Young Scholars No.232300421057 (Z.-Y.Z.), Henan Province Outstanding Young Talent Project in Health Science and Technology Innovation for Young and Middle-aged People YXKC2022035 (W.-W.W.), Henan Province Key Research and Development (R & D) program 232300421125 (W.-W.W.), the Science and Technology Program of Henan Province No.202102310136 (Y.-C.J.), 212102310113 (B.Y.), Guangdong Basic and Applied Basic Research Foundation 2020B1515120046 (Z.-C.L.), Guangdong Key Project 2018B030335001 (Z.-C.L.), Key Laboratory for Magnetic Resonance and Multimodality Imaging of Guangdong Pro-vince 2020B1212060051 (Z.-C.L.), and Guangzhou Key Research and Development Program 202007030002 (Z.-C.L.).

## Author contributions

Z.-Y.Z., W.-W.W., Z.-C.L., W.-C.L. designed and directed the research; Z.-Y.Z., W.-W.W., Z.-C.L., Y.-S.Z., L.-H.T., J.Y., H.-R.Z., D.L. processed the data, performed the experiments, and drafting of manuscript; Z.-C.L., Y.-S.Z., J.-X.D., and Q.-C.S. wrote and verified the code; L.-H.T., Yang G., Y.-N.Q., D.-L.P., L.W., W.-C.D., M.-K.W., S.-N.W., H.-L.D., C.S., Yu G., L.L., Z.-X.G., F.-Z.G., Z.-L.W., A.-Q.X., Z.-Y.L., H.-Y.Z., L.C., L.Z., B.Y. acquired the data and specimen; Z.-Y.Z., W.-W.W., J.Y., G.-Z.J., Y.-C.J., D.-M.Y., X.-Z.L., W.-C.L. verified the data; All authors have read and approved the final version of the manuscript.

## Competing interests
The authors declare no competing interests.

## Additional information

[1]Department of Pathology, The First Affiliated Hospital of Zhengzhou University, Zhengzhou, Henan, China. [2]Institute of Biomedical and Health Engineering, Shenzhen Institute of Advanced Technology, Chinese Academy of Sciences, Shenzhen, China. [3]University of Chinese Academy of Sciences, Beijing, China. [4]Department of Pathology, Xuanwu Hospital, Capital Medical University, Beijing, China. [5]Department of MRI, The First Affiliated Hospital of Zhengzhou University, Zhengzhou, Henan, China. [6]Department of Neurosurgery, Henan Provincial People's Hospital, Zhengzhou, Henan, China. [7]Department of Neurosurgery, The First Affiliated Hospital of Zhengzhou University, Zhengzhou, Henan, China. [8]The Key Laboratory of Biomedical Imaging Science and System, Chinese Academy of Sciences, Shenzhen, China. [9]National Innovation Center for Advanced Medical Devices, Shenzhen, China. [10]Present address: National Innovation Center for Advanced Medical Devices, Shenzhen, China. [11]These authors contributed equally: Weiwei Wang, Yuanshen Zhao, Lianghong Teng, Jing Yan. ✉e-mail: liwencai@zzu.edu.cn; zc.li@siat.ac.cn; fcczhangzy1@zzu.edu.cn

