## [Peer Review File · Nature Communications]

Neuropathologist-level integrated classification of adult-type diffuse gliomas using deep learning from whole-slide pathological imagesREVIEWER COMMENTS

Reviewer #1 (Remarks to the Author): Expert in digital pathology and machine learning

the authors use a deep learning method to subtype gliomas based on pathology images. This is an interesting application of a commonly used technology because the WHO glioma classification has recently changed. The main innovation of this study is the application to this new clinical problem.

Major

After looking at the code, it seems that the methods are generally valid. However, the quality of the code is suboptimal, it is not written according to good software engineering practice and seems to be very specific to this particular dataset. The pipeline is highly engineered and complex and I am not entirely sure about its reproducibility. Compared to other end-to-end methods, this seems to be very specific for this task. In its present form, it is not reusable for other similar problems. Also, the present form of the code is very verbatim and this has a risk of hidden (non-obvious) errors. The quality of the code could be much improved and I believe this would only be 1-2 days of work.

In general, I strongly believe that the authors should make the code available for all readers on GitHub and include a link in their manuscript. This is the standard in the field and almost all papers in the field of computational pathology do this, including recent paper in „Nature“-branded journals (e.g. <https://www.nature.com/articles/s41586-021-03512-4> or <https://www.nature.com/articles/s41591-022-01709-2.pdf?proof=t;>).

The study must adhere to relevant reporting guidelines and state the adherence to the guidelines. The Equator network should be checked for relevant guidelines (STARD, TRIPOD, etc.)

The statement "Nonetheless, patch-level methods still require pathologists or algorithms to judge the tissue type for each patch." is simply not true. Weakly supervised patch based methods are the standard in the field. Multiple clinically approved tools (DoMore, Owkin, Paige, Stratipath...) use this workflow. They report patient-level statistics and no one is checking every single patch.

Minor

In the abstract, the number of image patches is irrelevant, only the number of patients is relevant. The number in the train and test set should be mentioned in the abstract.

Previous work on weakly supervised computational pathology is not entirely acknowledged, more relevant citations from the last 1-2 years can be added to the introduction.

Reviewer #3 (Remarks to the Author): Expert in glioma imaging, radiology, and machine learning

Q1: Authors proposed an integrated diagnosis model for automatic classification of diffuse gliomas from annotation-free standard WSIs. The K-means method was used for patch clustering. However, the k-means clustering method can only converge to the local optimal, and the clustering effect is greatly affected by the initial value. Furthermore, authors used silhouette coefficient to determine the optimal clustering number. When the silhouette coefficient is low or negative, the clustering may be unstable, and the results may be unreliable. How did the author solve the problem of unstable clustering?

Q2: This study used private dataset from only two hospitals, the validity of the method should be validated in more external datasets or other public datasets.

Q3: There are many studies similar to the author's work, but we observed that the classification accuracy of this work is much higher than other works. Whether the difference arises from the dataset or the methodology, we would like to see the comparisons with more advanced methods.

In addition, the authors should discuss the reason why this cluster-based approach is superior to other approaches.

Q4: The sample class imbalance problem is also existed in this paper. In particular, the proportion of A3 and A4 samples is 2% and 3%, and that of GBM is more than 60%. Sample equalization processing is needed to make the results more convinced. During training, it is mentioned that the clinical parameters among training cohorts were balanced, however, it is better that the cohorts are randomly divided for several times, and the statistical distribution of the model evaluation results should be given to ensure the robustness and clinical usability of the proposed model.

Q5: The evaluation standard of multi-classification problem is different from that of binary classification, and the evaluation indicators used are also different. At least 3 of the 7 classification tasks mentioned in this paper are more than three-classification problems. The article seems to use multiple 1-to-N manners for evaluation, which is not clearly explained. In addition, if it is a 1-to-N manner, when A3 and A4 are classified as 1-to-N, the problem of balance of the data got worse (2% vs 98%, 3% vs 97%). The specific modeling process and evaluation criteria should be explained more clearly. Model and parameter details also need to be defined more clearly.

Q6: What is the meaning of the decimals less than 1 given in the confusion matrix? If it means proportion, the sum of rows and columns is not equal to 1.

Q7: It is mentioned in the method that the AUC comparison between models used the Delong test, but the comparison results (p value) are not provided in the results section.

Q8: The authors emphasized that developing an integrated diagnosis model directly from WSI to classify the types according to the newest 2021 WHO rule is in demand, however, the key biomarkers of GBM, including TERT promoter mutations, EGFR amplification and +7/-10 copy number changes, were not detected in this study, and all IDH-wild type gliomas were directly regarded as GBM in this study. This approximation might not be strict. IDH-wild cases without complete molecular diagnosis should be excluded

Q9 : How far is this AI-based diffuse gliomas classification technology from practical clinical practice, what its clinical limitations are, and what aspects of research can bring real breakthroughs.

Responses to the Reviewers

We thank the reviewers for their valuable comments. We have revised the manuscript according to these comments. Please note that all the changes made in the revised manuscript were shown with track changes. In the following, please find our point-to-point responses.

Reply to Reviewer 1

Reviewer #1:

The authors use a deep learning method to subtype gliomas based on pathology images. This is an interesting application of a commonly used technology because the WHO glioma classification has recently changed. The main innovation of this study is the application to this new clinical problem.

Comment 1 (Major): After looking at the code, it seems that the methods are generally valid. However, the quality of the code is suboptimal, it is not written according to good software engineering practice and seems to be very specific to this particular dataset. The pipeline is highly engineered and complex and I am not entirely sure about its reproducibility. Compared to other end-to-end methods, this seems to be very specific for this task. In its present form, it is not reusable for other similar problems. Also, the present form of the code is very verbatim and this has a risk of hidden (non-obvious) errors. The quality of the code could be much improved and I believe this would only be 1-2 days of work.

In general, I strongly believe that the authors should make the code available for all readers on GitHub and include a link in their manuscript. This is the standard in the field and almost all papers in the field of computational pathology do this, including recent paper in Nature-branded journals (e.g. <https://www.nature.com/articles/s41586-021-03512-4> or <https://www.nature.com/articles/s41591-022-01709-2.pdf?proof=t;>).

Response: Thanks very much for the reviewer's comment. We are sorry for the suboptimal quality of the code. According to the reviewer's comment, we have substantially improved the source code to enhance its readability and reproducibility. Firstly, we placed the fully functional code in separate script files with clear annotations, including the image clustering script, model training script, model testing script and so on. Secondly, we uploaded WSI patch files from six representative patients (classified as A2, A3, A4, O2, O3, and GBM, respectively) for model testing. Thirdly, we prepared a Readme file that included system requirement, installation guide, instructions for running the code on our test data.

In contrast to Github, another publicly available code repository, CodeOcean not only allows permanent storage of code and data for public access, but also provides an environment for running the code online. Therefore, we have uploaded the source code and a small test dataset to demo the code into CodeOcean. This is a convenient way for visitors to test the reproducibility of our results. By clicking the "Reproducible Run" button on the CodeOcean interface, visitors can run the Python script online without manual configuration and output the classification results of the test set patients. We have submitted the code and test data for publication on CodeOcean, and

it is currently undergoing verification, which means it is not yet publicly available via a link. The revised manuscript is being submitted concurrently with the submission of the code and test data to CodeOcean. Instead, we have uploaded the same code and test data as a single zip file (the Readme file was included in a folder named “code”) into a publicly available depository provided by Google Drive on at https://drive.google.com/file/d/1X1gXBg7nTP4Ulel_GBQ_fw-EMBLfTypz/view?usp=share_link. The editors and reviewers can access our code and test data via this link. After undergoing verification, the code and test data in CodeOcean will be made available with a DOI and a publicly accessible link. These will be included in the Code availability statement within the manuscript, replacing the current Google Drive link, following the completion of the review process.

Comment 2 (Major): The study must adhere to relevant reporting guidelines and state the adherence to the guidelines. The Equator network should be checked for relevant guidelines (STARD, TRIPOD, etc.)

Response: Thank the reviewer for the comment. We are sorry for not including this guideline in our original submission. The TRIPOD checklist is used for multivariable prediction, while STARD is made for reporting diagnostic accuracy studies. Therefore, in this pathology image-based diagnostic study, we chosen to use STARD (version 2015) checklist to assess whether all essential information has been included. We stated the adherence to the STARD guideline by adding one sentence in the last of subsection 2.4 Statistical analysis in the revised manuscript.

Comment 3 (Major): The statement "Nonetheless, patch-level methods still require pathologists or algorithms to judge the tissue type for each patch." is simply not true. Weakly supervised patch based methods are the standard in the field. Multiple clinically approved tools (DoMore, Owkin, Paige, Stratipath...) use this workflow. They report patient-level statistics and no one is checking every single patch.

Response: We thank the reviewer for pointing out the incorrect statement. We admit our statement is inaccurate. To avoid confusion, we deleted this statement in the revised manuscript.

Comment 4 (Minor): In the abstract, the number of image patches is irrelevant, only the number of patients is relevant. The number in the train and test set should be mentioned in the abstract.

Response: According to the reviewer’s suggestion, we have removed the number of image patches and mentioned the number of patients in the training, validation, and test sets in the revised abstract.

Comment 5 (Minor): Previous work on weakly supervised computational pathology is not entirely acknowledged, more relevant citations from the last 1-2 years can be added to the introduction.

Response: We are sorry for not entirely including relevant citations from the last 1-2 years. After carefully searching recent articles, we added 5 more relevant citations (Ref. [18-22]) to the introduction. All these articles focus on weakly supervised computational pathology. Specifically, the studies in references [18-21] presented weakly supervised computational pathology models for predicting molecular alterations, cardiac allograft rejection, and origins for cancers of unknown primary. While the study in reference [22] systematically compared six weakly supervised

methods in six clinically relevant tasks. The newly added articles are listed in the following:

[18] Schrammen PL, Laleh NG, Echle A, et al. Weakly supervised annotation-free cancer detection and prediction of genotype in routine histopathology. *Journal of Pathology*. 2022; 256(1):50-60.

[19] Bilal M, Raza SEAR, Azam A, et al. Development and validation of a weakly supervised deep learning framework to predict the status of molecular pathways and key mutations in colorectal cancer from routine histology images: a retrospective study. *The Lancet Digital Health*. 2021;3(12): e763-e772.

[20] Lipkova J, Chen TY, Lu MY, et al. Deep learning-enabled assessment of cardiac allograft rejection from endomyocardial biopsies. *Nature Medicine*. 2022; 28:575-582.

[21] Lu MY, Chen TY, Williamson DFK, et al. AI-based pathology predicts origins for cancers of unknown primary. *Nature*. 2021; 594:106-110.

[22] Laleh NG, Muti HS, Loeffler CML, et al. Benchmarking weakly-supervised deep learning pipelines for whole slide classification in computational pathology. *Medical Image Analysis*. 2022;79:102474.

Reply to Reviewer 3

Reviewer #3:

Comment 1: Authors proposed an integrated diagnosis model for automatically classification of diffuse gliomas from annotation-free standard WSIs. The K-means method was used for patch clustering. However, the k-means clustering method can only converge to the local optimal, and the clustering effect is greatly affected by the initial value. Furthermore, authors used silhouette coefficient to determine the optimal clustering number. When the silhouette coefficient is low or negative, the clustering may be unstable, and the results may be unreliable. How did the author solve the problem of unstable clustering?

Response: We thank the reviewer for this valuable comment. Yes, the sensitivity of K-means to initial values can lead to unstable clustering results. In some cases, if the initialization of cluster centers is not appropriate, K-means can result in arbitrarily bad clusters. To deal with this issue, there are several strategies that can be used to improve the stability, including multiple initialization (running K-means several times with different initialization), smart initialization (using selected initial centers instead of randomly sampled ones, such as the K-mean++ method), and data preprocessing. In our study, we used all three strategies. First, we applied a preprocessing pipeline on all WSIs, as described in subsection 2.3. Second, we used K-mean++ to improve the stability and reduce the chances of converging to local optima. In K-means++, the points that are farther away from the existing centers are more likely to be selected as the next center. This property is achieved by choosing the initial centers with a probability proportional to the squared distance from each point to the closest existing center. This ensures that the initial centers can be well spread out and are representative of the data. Third, we run K-means ten times with different sets of initial centers. The final result is the best output of the ten consecutive runs. In our study, K-means clustering was performed by using a KMeans function in the scikit-learn python library, with the function parameters `init = 'k-means++'` and `n_init = 10`. We have added two sentences in

Supplementary A5 to describe the K-means++ and multiple initialization approaches.

Yes, the silhouette coefficient measures how similar a data point is to its own cluster compared to other clusters. The silhouette coefficient ranges from -1 to 1, with higher values indicating better clustering. A low or negative silhouette coefficient can happen in the cases of noisy data, containing outliers, or when the clusters are too close together or too spread out. As described above, we improved the clustering stability from three aspects, which can reduce the chances of having bad clustering results with low or negative silhouette coefficients. In our study, we assessed different numbers of clusters ranging from 2 to 12, and the clustering number with the highest silhouette coefficient value was achieved when $K = 9$. We also calculated another metrics named Calinski-Harabasz index to assess the clustering quality. Calinski-Harabasz index measures the ratio of the between-cluster variance to the within-cluster variance, where a higher value indicates better clustering. At $K=9$ the Calinski-Harabasz index also reached its highest point, providing additional support for this optimal cluster number. We also added content describing the Calinski-Harabasz index in subsection 2.4, subsection 3.2, caption for **Figure 3**, and **Supplementary A5**. In the revised **Figure 3A**, the Calinski-Harabasz index was added.

Comment 2: This study used private dataset from only two hospitals, the validity of the method should be validated in more external datasets or other public datasets.

Response: Thanks very much for the reviewer's suggestion. As the reviewer suggested, we have included a relatively large sample of adult-type diffuse glioma dataset (external validation dataset 2, $n = 328$) from another top neurosurgery center (Xuanwu Hospital Capital Medical University, Beijing) in China. To ensure the concordance to previous WSI data, the same MAGSCAN-NER scanner (KF-PRO-005, KFBIO) was used to scan the slide from the newly added dataset (external validation dataset 2) to obtain the WSI. This another external dataset was also classified according to the integrated diagnosis pipeline strictly according to 2021WHO classifications of CNS tumors. To the best of our knowledge, there is no publicly available dataset with both WSI and corresponding 2021WHO classifications of gliomas due to the sophisticated classification pipeline of 2021WHO classifications of CNS tumors that require numerous advanced molecular tests. TCGA has limited numbers of WSIs of gliomas that were classified according to the 2016 and 2007 WHO classifications, which is not appropriate for our study. We would also like to mention to the reviewer that the collection of the new external data, as well as performing a vast amount of molecular tests took much time for us, which caused a delayed revision. We are sorry for the delay. To our knowledge, the current study, which comprises of a training cohort ($n = 1362$), a validation cohort ($n = 340$), an internal testing cohort ($n = 289$), an external testing cohort 1 ($n = 305$), and an external testing cohort 2 ($n = 328$), has been by far the largest dataset of WSIs with complete molecular and histological information required for a strict classification of adult-type diffuse glioma according to the 2021WHO rule, and is the first study that utilize deep learning techniques for predicting the 2021 WHO integrated classifications of adult-type diffuse glioma from WSIs. We have modified the dataset description in subsection 2.1 and **Supplementary A1** in the revised submission.

Comment 3: There are many studies similar to the author's work, but we observed that the classification accuracy of this work is much higher than other works. Whether the difference arises from the dataset or the methodology, we would like to see the comparisons with more advanced

methods.

In addition, the authors should discuss the reason why this cluster-based approach is superior to other approaches.

Response: Thanks very much for the reviewer’s suggestion. Yes, there are many methods for computational pathology, e.g., MIL and its variants such as Attention-based MIL (AMIL) [1] and clustering-constrained-attention MIL (CLAM) [2], graph-based models, and self-supervised representation learnings methods. Selecting an optimal method is not easy. We would like to answer the reviewer’s questions from the following aspects.

(1) Previous studies: Although there have been numerous computational pathology studies, there is a lack of focus on the specific clinical task of diagnosing gliomas. We searched the PubMed database using the term “(deep learning) AND ((digital histopathology) OR (slide)) AND (glioma)” from 2018 to 2023, generating only 30 results. By further selecting studies focusing on diagnosis, and adding an early study published in 2015 and a machine learning study, we only found four studies in this specific task, as listed in Refs. [11,12,26,27] in the originally submitted manuscript ([11,12, 29,30] in the revised manuscript). The studies in [11,29,30] predicted only histologic grades according to the 2007 WHO classification. So far, we have found only one study aimed to classify five major categories considering both histological grades and molecular makers according to 2007/2016 WHO classifications [12]. They employed a classical weakly supervised method with an attention based DenseNet, where a fixed number of patches (300) were selected from each WSI for model building. Despite its simplicity, this model achieved a good accuracy of 87.5% in classifying the five categories. To date there are no diagnostic models for the 2021 WHO classification. This highlights the need for more studies that specifically address the clinical task of glioma diagnosis, particularly with regards to integrated diagnosis from histology images. Yes, the accuracy of our model was numerically higher. However, strictly speaking, we cannot directly compare our model with previous ones, as we are doing different classification tasks. Our model is the first one that classified gliomas into six types strictly adhering to the 2021 WHO rule.

Task	Types/Grades	proposed	MIL	AMIL	CLAM	All-patch
1	A2	0.959	0.920	0.920	0.924	0.901
		0.970	0.913	0.915	0.910	0.843
		0.934	0.903	0.906	0.919	0.877
		0.945	0.975	0.945	0.928	0.925
	A3	0.995	0.865	0.874	0.886	0.897
		0.973	0.918	0.833	0.900	0.747
		0.923	0.933	0.917	0.959	0.755
		0.944	0.856	0.856	0.863	0.868
	A4	0.953	0.895	0.933	0.935	0.863
		0.994	0.793	0.930	0.918	0.833
		0.987	0.997	0.852	0.871	0.893
		0.904	0.892	0.844	0.890	0.895
	O2	0.978	0.956	0.923	0.928	0.912
		0.932	0.916	0.926	0.938	0.856
		0.964	0.981	0.927	0.952	0.869
		0.942	0.886	0.910	0.923	0.906
	O3	0.982	0.948	0.899	0.896	0.947
		0.980	0.945	0.925	0.899	0.909
		0.978	0.950	0.886	0.927	0.886
		0.950	0.901	0.930	0.916	0.763
GBM	0.960	0.942	0.936	0.938	0.914	
	0.980	0.953	0.949	0.951	0.910	
	0.984	0.963	0.955	0.962	0.922	
	0.952	0.925	0.937	0.938	0.928	
2	A	0.961	0.909	0.885	0.894	0.899
		0.969	0.894	0.918	0.920	0.881

		0.938	0.897	0.932	0.930	0.846
		0.941	0.920	0.913	0.905	0.899
	O	0.974	0.960	0.921	0.955	0.919
		0.974	0.914	0.930	0.930	0.873
		0.973	0.981	0.915	0.962	0.892
		0.938	0.903	0.927	0.910	0.854
		0.960	0.942	0.936	0.938	0.914
	GBM	0.980	0.953	0.949	0.951	0.910
		0.983	0.963	0.955	0.962	0.922
		0.952	0.925	0.937	0.938	0.928

(2) Comparison with advanced models: When designing the study, we aimed to find an accurate weakly supervised method. We have tried classical MIL and several MIL variants. However, we found MIL and its attention-based versions were not that effective in our task. Finally, we found that an alternative method – the clustering-based model worked well for our task. In our original manuscript, we have reported the results of the classical MIL only. According to the reviewer’s comment, we showed the comparison of our method with two more advanced models, i.e., AMIL and CLAM. For both AMIL and CLAM, features were extracted using a pretrained ResNet-50 from each patch. Adam optimizer was used for both models, with a learning rate of 0.001 and a batch size of 64. Both models were trained for classifying the six types. The AUCs of all models in classifying the six types with grade (A2, A3, A4, O2, O3 and GBM) and in classifying the three major types (A, O, GBM) are summarized in the above table, where the AUCs on internal validation set, internal testing set, external testing set1, and external testing set 2 are shown for each type from top to bottom. This table with corresponding ROC curves were also submitted as **Table S4** and **Figure S11-S12**, respectively. We also added the content of comparison with AMIL and CLAM into the Materials and methods subsection 2.4 and Results subsection 3.4.

(3) Result analysis: The performance of the proposed cluster-based method was better than or comparable to CLAM, AMIL, and MIL. On the other hand, we didn’t find significant difference in performance among the three MIL models. In our tasks, the attention-based methods did not seem to bring much benefit as expected. One reason might be the high degree of variability and complexity within the pathologic data of glioma, making it hard to learn effective attention weights for instances related to the target classes. Specifically, to accurately classify the six types of A2, A3, A4, O2, O3 and GBM, we first need to identify discriminative pathologic morphology related to cancer types (A, O, and GBM) and grades within types (A2/3/4, O2/3). Furthermore, we need to recognize tumor genotypes with shared histological features (e.g., IDH-wildtype and -mutant tumors), which may be reflected by not only cell morphology but also unseen imaging patterns. Moreover, several key instance might be sparse (microvascular proliferation or necrosis). These key features might be contained in the same instances, or in many different instances, or in sparse instances. All these instances are important to the classification tasks, which are too diverse and complex to learn by an attention mechanism. On the other hand, we guess that the label noise induced by the simplified slide-to-patch label assignment would also impair the attention weights to some extent. All these factors motivated us to design an alternative approach with a different patch-sampling strategy. To address this complex classification problem, we considered that a direct strategy could potentially offer a solution. We abandoned the learning-based attention approach but used a straightforward way. We reconsidered the imaging patterns as “phenotypes”, which may reflect the cell morphology or/and underlying genetic alterations. We hypothesized that a group of patches might share similar phenotypes. Then, the goal is clear: finding out the patches with the most discriminative phenotypes. Since it is difficult to find individual key patches, we

might as well turn to searching for some important clusters of patches. An individual patches within a cluster might not be that important, but the entire cluster with similar phenotypes may be useful. To do so, we used the all-patch model as a benchmark, and selected the clusters with better discriminative power than the all-patch model. Then, we built the final model by using the patches within all selected clusters. Our data demonstrated that the cluster-based methods achieved good performance in the integrated diagnostic task. On the other hand, the large amount of data collected from multiple hospitals should be another important reason for the accuracy of our model, where all WSI slides from three hospitals were scanned using the same scanner to ensure consistency. The discussion of the reason why the cluster-based method performed better was added in the fifth paragraph of the Discussion section in the revised manuscript.

Comment 4: The sample class imbalance problem is also existed in this paper. In particular, the proportion of A3 and A4 samples is 2% and 3%, and that of GBM is more than 60%. Sample equalization processing is needed to make the results more convinced. During training, it is mentioned that the clinical parameters among training cohorts were balanced, however, it is better that the cohorts are randomly divided for several times, and the statistical distribution of the model evaluation results should be given to ensure the robustness and clinical usability of the proposed model.

Response: Yes, the classes are imbalance, which is mainly due to the variable incidence rate across the six glioma types included in this study. Our study recruited 2624 consecutive patients from three hospitals, and all patients had completed histologic and molecular information required for the strict 2021 WHO classification. Therefore, our data can well represent the prevalence of the 2021 WHO types of adult diffuse glioma. The incidence rate of the tumors in our study is basically in line with the data reported in previous studies [1-3].

The imbalance problem in our study is obvious, and we have noticed it in the stage of model development. We have applied data oversampling to augment the minority classes in our original model. In the originally submitted **Supplementary A7**, we have mentioned the data augmentation in the second paragraph. We are sorry for not describing it clearly. Specifically, to accommodate the class imbalance, we used random rotation and shear approaches to augment A3, A4, and O3 classes in the training dataset (after training/validation dataset division). The case number in each of the three classes have tripled after augmentation. More details of the augmentation operation have been added in the revised **Supplementary A6**. Our results (especially the F1-score and the newly added PR curves, as described in the response to Comment 5) demonstrated the stability of the model performance on the imbalanced data.

According to the reviewer's comment, we repeated the model development procedure in a five-fold cross-validation way, re-assigning patients into non-overlapping training and validation cohorts five times. The aims were to demonstrate the model robustness on different folds, and to select an optimal model among all folds. Please note that in our study, the internal testing cohort was generated by using a time-point division approach (patient cases after January 2020 were used as internal testing cohort while patients before January 2020 were randomly divided into training and validation cohorts, as described in subsection 2.1), which ensured the independence of the internal testing cohort. So, we performed five-fold cross-validation only on training and validation datasets. Considering the class imbalance, in each repetition the training and validation sets were generated using stratified random sampling at a ratio of 4:1, with patient characteristic balanced in

both sets, which was the same as used in our original modeling process. For each fold, the same data augmentation was used on A3, A4, and O3 classes in the training cohort, and the model with the lowest validation loss after a minimum of 50 training epochs was saved. Otherwise, the model was trained up to a maximum of 150 epochs. The entire procedure was tedious and took much time to train all the models on large datasets (hundreds of thousands of patches). The ROC curves for each fold and the mean ROC curves were plotted on both training and validation cohorts, respectively, as shown in **Figure S5**. The boxplots of AUCs for all folds were also plotted to show the statistical distributions, as shown in **Figure S6**. The results demonstrated that the model performance was stable across the folds. The large dataset used in the training stage should be one of the main reasons for the model stability reflected in the cross-validation results. Finally, we selected the optimal model among the five folds as the proposed model and tested its performance on the internal testing cohort and two external testing cohorts. We would like to mention that, the stable performance on the three independent testing datasets collected from multiple hospitals also provided an even stronger demonstration of the model robustness. We have added the content of the cross-validation method in subsections 2.1, 2.3, and 2.4 in the main text and in **Supplementary A6**, and reported the results in cross-validation in subsection 3.3, **Figure S5**, and **Figure S6** in the revised manuscript.

[1] JE Eckel-Passow, DH Lachance, AM Molinaro, et al. Glioma groups based on 1p/19q, IDH, and TERT promoter mutations in tumors. *The New England Journal of Medicine*. 2015; 372(26): 2499-2508.

[2] FP Barthel, KC Johnson, FS Varn, et al. Longitudinal molecular trajectories of diffuse glioma in adults. *Nature*. 2019; 576:112-120.

[3] QT Ostrom, M Price, C Neff, et al. CBTRUS Statistical report: Primary brain and other central nervous system tumors diagnosed in the United States in 2015-2019. *Neuro-Oncology*. 2022; 24(S5), v1-v95.

Comment 5: The evaluation standard of multi-classification problem is different from that of binary classification, and the evaluation indicators used are also different. At least 3 of the 7 classification tasks mentioned in this paper are more than three-classification problems. The article seems to use multiple 1-to-N manners for evaluation, which is not clearly explained. In addition, if it is a 1-to-N manner, when A3 and A4 are classified as 1-to-N, the problem of balance of the data got worse (2% vs 98%, 3% vs 97%). The specific modeling process and evaluation criteria should be explained more clearly. Model and parameter details also need to be defined more clearly.

Response: Yes, we used 1-to-N manner (also referred to as one-vs-rest) for evaluation of the models. The major reason for using a 1-to-N approach in our multi-class problems is that it allows for the evaluation of each class individually, which can provide more detailed insights into the strengths and weaknesses of the model for each class. The 1-to-N manners are also used in other imaging-based multi-class diagnosis problems, such as the glioma classification study in Ref. [12]. We have clarified this in subsection 2.4 Statistical analysis.

Yes, to well assess the model performance in this data imbalance problem, we also calculated the confusion matrix of the patch-level model, F1-score (already existed in our original manuscript, shown in **Figure S5** and **Table 1**, respectively), and the Precision-Recall (PR) curves of the patient-level model (newly added in the revised manuscript, shown in **Figure S7**). Specifically, the

confusion matrix provides detailed information about the number or proportion of true positives, true negatives, false positives, and false negatives for each class, which is more comprehensive than other any single metric such as AUC, and therefore is useful for data imbalance problems. The PR curve is often more useful than the ROC curve in a data imbalance problem because it can highlight the performance of the classifier on the minority class: a high precision score indicates that the classifier is making few false positive predictions (minority class), while a high recall score indicates that the classifier is correctly identifying a large proportion of positive samples (minority classes). F1 score is the harmonic mean of precision and recall. It balances both precision and recall and is often used as a single metric to evaluate the overall performance of a model, which is also more useful in data imbalance problems.

To address the data imbalance problem, we used data augmentation for minority classes, and applied five-fold cross-validation to select a more robust and optimal model, as described in detail in the responses to reviewer's comment 4.

Comment 6: What is the meaning of the decimals less than 1 given in the confusion matrix? If it means proportion, the sum of rows and columns is not equal to 1.

Response: Yes, the decimals in the confusion matrix mean proportion of each type. Yes, we found there were errors in the confusion matrix in our original manuscript, as the sum of columns was not equal to 1. We are sorry for this calculation mistake and appreciate the reviewer for pointing out this mistake. In our revised manuscript, the model was rebuilt, and the confusion matrix was updated, as shown in **Figure S5**.

We would like to mention the advantages of using decimals in our confusion matrix. First, compared with whole numbers, decimals allow us to quickly understand the proportion of correct and incorrect predictions made by the model, regardless of the size of the dataset. Second, decimals can be particularly useful when dealing with datasets with imbalanced classes. Third, it can help to highlight the important aspects of the matrix, e.g., the elements in the diagonal line were highlighted in our study. If using whole numbers instead, the elements with the large number would be highlighted, which may not well reflect insights and conclusions from the results.

Please note that the sum of the ground truth proportion for each tumor type shown in one column is equal to 1. However, the sum of rows, which represent the prediction made by the model, might not be equal to 1, because there are instances that have been incorrectly predicted by the model. For example, if an instance belongs to class A, but the model predicts it as belonging to class B, then this instance will be counted in the row corresponding to class B, leading to a sum greater than 1 in that row, and a sum less than 1 in the row corresponding to class A.

Comment 7: It is mentioned in the method that the AUC comparison between models used the Delong test, but the comparison results (p value) are not provided in the results section.

Response: We are sorry for neglecting to report the Delong p values. We compared the AUC of our proposed clustering-based model with that of four other models in our study - the classical MIL model, the attention-based MIL (AMIL) model, the clustering-constraint-attention MIL (CLAM) model, and the all-patch model - using the Delong test. The resulted P values were summarized in **Table S5**, which was added into the revised submission.

For classifying the six types of A2, A3, A4, O2, O3, and GBM (task 1), most P values for the three MIL models were more than 0.05, indicating that the difference in AUCs between the clustering-

based model and the MIL models was not significant on most datasets. However, in most tests the AUCs of our proposed model were numerically higher than that of the MIL models. Meanwhile, we found many P values less than 0.05 for the all-patch model, indicating that the AUCs between the proposed model and the all-patch model were significantly different in many tests. For classifying the three types of A, O, and GBM (task 2), P values in almost half of tests were less than 0.05, where P values were less than 0.05 for the all-patch model in all tests. We have added the results of Delong tests in **Table S5**, and described the results in brief in subsection 3.4 of the revised manuscript.

Comment 8: The authors emphasized that developing an integrated diagnosis model directly from WSI to classify the types according to the newest 2021 WHO rule is in demand, however, the key biomarkers of GBM, including TERT promoter mutations, EGFR amplification and +7/-10 copy number changes, were not detected in this study, and all IDH-wild type gliomas were directly regarded as GBM in this study. This approximation might not be strict. IDH-wild cases without complete molecular diagnosis should be excluded

Response: Thanks very much for the reviewer's comment. For more precise and strict classification according to the 2021 WHO rule, we have reselected near 4 hundreds of formalin-fixed, paraffin embedded (FFPE) tissues from patients with grade 2/3 IDH-wildtype diffuse astrocytic gliomas, and performed near 8 hundreds molecular tests including Sanger sequencing for TERT promoter mutations, fluorescence in situ hybridization (FISH) for EGFR amplifications and chromosome 7 gain /10 loss, for determining grade 2/3 IDH-wildtype diffuse astrocytic gliomas with TERT promoter mutations, or EGFR amplification or +7/-10 copy number changes as Glioblastoma, IDH-wildtype (GBM). The detailed protocols of the multiple molecular tests are described in **Supplementary A2-A3**, and representative results of IDH1/IDH2 mutations, TERT promoter mutations, Chromosome 1p/19q co-deletions, CDKN2A homozygous deletion, EGFR amplification and Chromosome 7 gain/chromosome 10 loss are depicted in **Figure S1-4**. The integrated classification pipeline precisely according to 2021 WHO Classification of adult-type diffuse gliomas in the current study was shown in **Figure 2** and described in **Supplementary A4**.

In the FAZZU data set, there were 306 cases of grade 2/3 IDH-wildtype diffuse astrocytic gliomas, in which FFPE tissues in 29 cases were not sufficient to complete necessary molecular tests. In the remaining 277 cases, 157 cases were detected as TERT promoter mutations, 2 remaining cases were detected as EGFR amplification, and 1 remaining case was detected as chromosome 7 gain /10 loss. Therefore, there were 160 grade 2/3 IDH-wildtype diffuse astrocytic gliomas diagnosed as GBM, and the other 117 cases without TERT promoter mutations, nor EGFR amplification, nor +7/-10 copy number changes were excluded. In the HPPH data set (external testing cohort 1), there are 49 cases of grade 2/3 IDH-wildtype diffuse astrocytic gliomas, in which FFPE tissues in 2 cases were not sufficient to complete necessary molecular tests. In the remaining 47 cases, 29 cases were detected as TERT promoter mutations, 1 remaining case was detected as EGFR amplification, and no remaining case was detected as chromosome 7 gain /10 loss. Therefore, there were 30 grade 2/3 IDH-wildtype diffuse astrocytic gliomas diagnosed as GBM, and the other 17 cases without TERT promoter mutations, nor EGFR amplification, nor +7/-10 copy number changes were excluded. In the XHCMU data set (external testing cohort 2), there were 43 cases of grade 2/3 IDH-wildtype diffuse astrocytic gliomas, in which FFPE tissues in 1 case were not sufficient to complete necessary molecular tests. In the remaining 42 cases, 25 cases were detected

as TERT promoter mutations, 1 remaining case was detected as EGFR amplification, and 1 remaining case was detected as chromosome 7 gain /10 loss. Therefore, there were 27 grade 2/3 IDH-wildtype diffuse astrocytic gliomas diagnosed as GBM, and the other 15 cases without TERT promoter mutations, nor EGFR amplification, nor +7/-10 copy number changes were excluded. After the molecular test, all GBM were precisely classified according to the 2021 WHO rule. Finally, in total we had 1991 patients from FAZZU and 305 patients from HPPH. For patients from FAZZU, 289 patients collected between January 2020 and December 2020 were used as an internal testing cohort, while 1702 patients from January 2011 to December 2019 were used for model training and validation. We would like to mention that in our original division, 1464 patients were randomly assigned into the training cohort while 366 patients were assigned into the validation cohort, at a ratio of 4:1. After testing the three markers of TERT, EGFR and +7/-10, 1382 patients remained in training cohort while 320 patients remained in validation cohort. The ratio changed to 4.32:1, which was not commonly used in deep learning model development, could cause confusion for readers if preserved. Therefore, we randomly re-assigned the 1702 patients from January 2011 to December 2019, resulting in a new training cohort of 1362 patients and a validation cohort of 340 patients, at a more commonly used ratio of 4:1. As a result of this change of the training data, we re-trained the entire model using the new data division. In addition, we implemented five-fold cross-validation based on the 1702 patients in response to Comment 4 from the reviewer. This approach allowed us to validate the robustness of the model and identify the optimal version.

Comment 9: How far is this AI-based diffuse gliomas classification technology from practical clinical practice, what its clinical limitations are, and what aspects of research can bring real breakthroughs.

Response: Thanks very much for the reviewer's comments. We think the AI-based diffuse gliomas classification technology is now rapidly approaching to the goal of application in real clinical scenarios. However, despite of promising results in research studies, there are still several challenges that need to be addressed before it can be widely implemented in clinical practice.

1) One of the major limitations is the lack of WSI data with precise diagnostic labels used for model training. This can lead to overfitting and poor generalization to new datasets. While having more data is generally beneficial, it is unclear how many WSIs are needed to train a robust model. Furthermore, the performance of AI models can be impacted by factors such as variable acquisition protocols and image quality. In our study, all WSI slides from three hospitals were scanned using the same digital scanner to ensure consistency. To address the impact of scanner variability and develop a classifier with good robustness in clinical practice, we plan to collect a large number of WSIs obtained from a variety of scanners in the future. We have added this in limitations in our revised manuscript.

2) AI models may provide accurate predictions, but the data-driven nature makes them more like black-boxes, which means that it can be difficult to understand why and how the model arrived at its predictions. This can make it challenging for clinicians to integrate the predictions into clinical decision-making. The meaning of the features extracted from the images should be explained in a more transparent way. In addition, the biologic or biomedical meanings of the features should also be elucidated, enhancing the clinician's confidence when using the AI classifiers. For example, it would be better if the model can make a prediction of the tumor types and can further tell the key

cell morphology/molecular alterations like a histopathologist/molecular pathologist.

3) There are many regulatory challenges and ethical concerns, such as issues related to patient privacy, transparency, and accountability.

To achieve real breakthroughs in the field of AI-based glioma classification just like the “chatGPT time”, two key factors are crucial in our opinion: a massive amount of labeled WSI data and significant computational power. We estimate that over a million labeled WSIs obtained from diverse scanners across multiple countries would be necessary to train a robust model. Additionally, more efficient unsupervised or weakly supervised representation learning methods should be developed to extract features from the WSIs. WSIs without diagnostic labels can also be collected to develop a powerful representation learning method. One possible approach is to divide the glioma classification problem into several smaller downstream tasks, such as type classification, grade classification, IDH mutation recognition, 1p/19q codeletion recognition and so on. The powerful features representations can be used to in these downstream tasks. Although promising, the real breakthroughs are not easy. Current study presented a research model using a large WSI dataset with precise 2021 WHO classifications, making a step forward clinically accepted AI models.

REVIEWERS' COMMENTS

Reviewer #1 (Remarks to the Author):

The authors have addressed all of my comments. No further requests.

Reviewer #4 (Remarks to the Author): Expert in glioma imaging and AI; replaces Reviewer #3

By reading the enhanced version of the manuscript, the reviewers' comments and the authors' replies, I believe that the authors have sufficiently demonstrated the strength, efficacy, and reproducibility of their work.

I believe that it will be a great paper to be added to the current literature regarding the application of deep learning and machine learning to digital pathology.

Some typos and mistakes should still be addressed in the manuscript, such as awkward definitions like "in histology" at line 82, or typos like "microvascular prefiltration" at line 368, or "k-means" in figure 1, step 1 (patch clustering).

Moreover, the analysis of the existing literature is still limited, with many articles in the field not quoted here, e.g., Jose L et al on AI-assisted classification of gliomas using WSI (Arch Pathol Lab Med 2022), and, more importantly, the use of DL and generative-AI (GAN) to predict IDH status in histological images of glioma (Liu S et al, Sci Rep 2020), or weakly supervised learning for the same task (Ma et al., J Neurooncol 2023). A more comprehensive literature review would make this paper more convincing.

Some comments on the stain normalisation also would strengthen the paper, i.e., how different stain intensities may have affected DL-analysis.

Responses to the Reviewers

We thank the reviewers for their valuable comments. We have revised the manuscript according to these comments. Please note that all the changes made in the revised manuscript were shown with track changes. In the following, please find our point-to-point responses.

Reply to Reviewer 1

Reviewer #1:

Comment 1: The authors have addressed all of my comments. No further requests.

Response: Thanks very much for the reviewer's valuable comments.

Reply to Reviewer 4

Reviewer #4

Comment 1: By reading the enhanced version of the manuscript, the reviewers' comments and the authors' replies, I believe that the authors have sufficiently demonstrated the strength, efficacy, and reproducibility of their work.

I believe that it will be a great paper to be added to the current literature regarding the application of deep learning and machine learning to digital pathology.

Some typos and mistakes should still be addressed in the manuscript, such as awkward definitions like "in histology" at line 82, or typos like "microvascular prefiltration" at line 368, or "k-means" in figure 1, step 1 (patch clustering).

Response: We thank the reviewer for pointing out the typos and mistakes. We have reexamined and modified the typos and mistakes all through the revised manuscript, including the ones pointed out by the reviewer.

Comment 2: Moreover, the analysis of the existing literature is still limited, with many articles in the field not quoted here, e.g., Jose L et al on AI-assisted classification of gliomas using WSI (Arch Pathol Lab Med 2022), and, more importantly, the use of DL and generative-AI (GAN) to predict IDH status in histological images of glioma (Liu S et al, Sci Rep 2020), or weakly supervised learning for the same task (Ma et al., J Neurooncol 2023). A more comprehensive literature review would make this paper more convincing.

Response: According to the reviewer's suggestion, we have added the three articles as Refs [13-15] in our revised manuscript and cited them in the third paragraph of Introduction section and the third paragraph of Discussion section.

Comment 3: Some comments on the stain normalisation also would strengthen the paper, i.e., how different stain intensities may have affected DL-analysis.

Response: We thank the reviewer for this comment. As suggested by the reviewer, we have added comments on the stain normalization in the limitation paragraph in Discussion section in the revised manuscript.